**Registered Report**

# Negativity drives online news consumption

**Claire E. Robertson** [1,7]**, Nicolas Pröllochs** [2,7]**, Kaoru Schwarzenegger**[3]**, Philip Pärnamets** [4]**, Jay J. Van Bavel** [1,5] ✉ **& Stefan Feuerriegel** [3,6] ✉

Online media is important for society in informing and shaping opinions, hence raising the question of what drives online news consumption. Here we analyse the causal effect of negative and emotional words on news consumption using a large online dataset of viral news stories. Specifically, we conducted our analyses using a series of randomized controlled trials (N = 22,743). Our dataset comprises ~105,000 different variations of news stories from Upworthy.com that generated ~5.7 million clicks across more than 370 million overall impressions. Although positive words were slightly more prevalent than negative words, we found that negative words in news headlines increased consumption rates (and positive words decreased consumption rates). For a headline of average length, each additional negative word increased the click-through rate by 2.3%. Our results contribute to a better understanding of why users engage with online media.

The newsroom phrase 'if it bleeds, it leads' was coined to reflect the intuition among journalists that stories about crime, bloodshed and tragedy sell more newspapers than stories about good news[1]. However, a large portion of news readership now occurs online—the motivation to sell papers transformed into a motivation to keep readers clicking on new articles. In the United States, 89% of adults get at least some of their news online, and reliance on the Internet as a news source is increasing[2]. Even so, most users spend less than 5 minutes per month on all of the top 25 news sites put together[3]. Hence, online media is forced to compete for the extremely limited resource of reader attention[4].

With the advent of the Internet, online media has become a widespread source of information and, subsequently, opinion formation[5–9]. As such, online media has a profound impact on society across domains such as marketing[10,11], finance[12–14], health[15] and politics[16–19]. Therefore, it is crucial to understand exactly what drives online news consumption. Previous work has posited that competition pushes news sources to publish 'click-bait' news stories, often categorized by outrageous, upsetting and negative headlines[20–22]. Here we analyse the effect of negative words on news consumption using a massive online dataset of viral news stories from Upworthy.com—a website that was one of the most successful pioneers of click-bait in the history of the Internet[23].

The tendency for individuals to attend to negative news reflects something foundational about human cognition—that humans preferentially attend to negative stimuli across many domains[24,25]. Attentional biases towards negative stimuli begin in infancy[26] and persist into adulthood as a fast and automatic response[27]. Furthermore, negative information may be more 'sticky' in our brains; people weigh negative information more heavily than positive information, when learning about themselves, learning about others and making decisions[28–30]. This may be due to negative information automatically activating threat responses—knowing about possible negative outcomes allows for planning and avoidance of potentially harmful or painful experiences[31–33].

Previous work has explored the role of negativity in driving online behaviour. In particular, negative language in online content has been linked to user engagement, that is, sharing activities[22,34–39]. As such, negativity embedded in online content explains the speed and virality of online diffusion dynamics (for example, response time, branching of online cascades)[7,34,35,37,39–41]. Further, online stories from social media perceived as negative garner more reactions (for example, likes, Facebook reactions)[42,43]. Negativity in news increases physiological activations[44], and negative news is more likely to be remembered by users[45–47]. Some previous works have also investigated negativity

---

[1]Department of Psychology, New York University, New York, NY, USA. [2]Department of Business and Economics, University of Giessen, Giessen, Germany. [3]Department of Management, Technology and Economics, ETH Zurich, Zurich, Switzerland. [4]Division of Psychology, Department of Clinical Neuroscience, Karolinska Institutet, Stockholm, Sweden. [5]Center for Neural Science, New York University, New York, NY, USA. [6]LMU Munich School of Management, LMU Munich, Munich, Germany. [7]These authors contributed equally: Claire E. Robertson, Nicolas Pröllochs. ✉e-mail: jay.vanbavel@nyu.edu; feuerriegel@lmu.de

## Table 1 | Design table

| Question | Hypothesis | Sampling plan | Analysis plan | Interpretation given to different outcomes |
|---|---|---|---|---|
| How does the presence of negative and positive words affect the CTR for news headlines? | The presence of negative words in a headline will significantly increase the CTR for that headline. The presence of positive words in a headline will significantly decrease the CTR for that headline. | A power analysis suggested that the sample size of the confirmatory dataset (22, 743 RCTs) will have sufficient power to achieve 99% power to detect an effect size of 0.01, considered to be a small effect size[82]. This effect size is slightly more conservative than estimates of effect sizes from pilot studies and is derived from theory[76]. | We conducted a multilevel binomial model examining the effects of the proportion of negative words in a headline on the CTR, adjusting for the proportion of negative words in a headline, the number of positive words in the headline, the complexity of the headline as measured by the Gunning Fog Index and the age of the story relative to the age of the Upworthy platform. We included random effects grouped by RCT and used two models to test our hypothesis. One allows the intercept to vary, the other also allows the slope of negative words to vary. | A significant positive coefficient for negative words is interpreted as evidence that a higher ratio of negative words in a headline is associated with a greater CTR. A significant negative coefficient for negative words is interpreted as evidence that a higher ratio of negative words in a headline is associated with a lower CTR. A significant positive coefficient for positive words is interpreted as evidence that a higher ratio of positive words in a headline is associated with a greater CTR. A significant negative coefficient for positive words is interpreted as evidence that a higher ratio of positive words in a headline is associated with a lower CTR. We consider evidence to be conclusive only in cases where both model fits to the data agree in their qualitative conclusions about the effect of negative words. <br><br>To evaluate effects where we cannot reject the null hypothesis, we will test for equivalence[67] against an interval of (−0.001, 0.001). If our observed confidence interval is fully contained in this interval, we will consider this as evidence for a null effect, otherwise we will consider the results inconclusive with respect to the null. |
| How does the presence of discrete emotional words affect the CTR for news headlines? | The presence of anger, fear and sadness words in a headline will significantly increase the CTR for that headline. The presence of joy words in a headline will significantly decrease the CTR for that headline. | A power analysis suggested that the sample size of the confirmatory dataset (22,743 RCTs) will have sufficient power to achieve 99% power to detect effect sizes of 0.01. | We conducted a multilevel binomial model examining the effects of the four emotions (anger, fear, joy and sadness) on the CTR, adjusting for the number of words in the headline, the complexity of the headline as measured by the Gunning Fog Index and the age of the story relative to the age of the Upworthy platform. We included the RCT as a random intercept. | A positive value for each of the emotions signifies a larger proportion of emotional words from that emotion in a headline. Therefore, a significant positive coefficient for the emotion is interpreted as evidence that headlines containing a word from the emotion (that is, anger, fear, joy and sadness) are associated with a greater CTR. Conversely, a negative value for each of the coefficients signifies that the proportion of emotional words from the emotion is more prevalent in a headline. Therefore, a significant negative coefficient for the emotion indicates that headlines containing a word from the emotion (that is, anger, fear, joy and sadness) are associated with a smaller CTR. <br><br>To evaluate effects where we cannot reject the null hypothesis, we tested for equivalence[67] against an interval of (−0.001, 0.001). If our observed confidence interval is fully contained in this interval, we will consider this as evidence for a null effect, otherwise we will consider the results inconclusive with respect to the null. |

effects for specific topics such as political communication and economics[34,48–52]. Informed by this, we hypothesized an effect of negative words on online news consumption.

The majority of studies on online behaviour are correlational[34–36,38–42], while laboratory studies take subjects out of their natural environment. As such, there is little work examining the causal impact of negative language on real-world news consumption. Here we analyse data from the Upworthy Research Archive[53], a repository of news consumption data that are both applied and causal. Due to the structure of this dataset, we are able to test the causal impact of negative (and positive) language on news engagement in an ecologically rich online context. Moreover, our dataset is large-scale, allowing for a precise estimate of the effect size of negative words on news consumption.

Data on online news consumption was obtained from Upworthy, a highly influential media website founded in 2012 that used viral techniques to promote news articles across social media[53,54]. Upworthy has been regarded as one of the fastest-growing media companies worldwide[53] and, at its peak, reached more users than established publishers such as the New York Times[55]. Content was optimized with respect to user responses through data-driven methods, specifically randomized controlled trials (RCTs)[56]. The content optimization by Upworthy profoundly impacted the media landscape (for example, algorithmic policies were introduced by Facebook in response)[23]. In particular, the strategies employed by Upworthy have also informed other content creators and news agencies.

Upworthy conducted numerous RCTs of news headlines on its website to evaluate the efficacy of differently worded headlines in generating article views[53]. In each experiment, Upworthy users were randomly shown different headline variations for a news story, and user responses were recorded and compared. Editors were commonly required to propose 25 different headlines from which the most promising headlines were selected for experimental testing[57].

In the current paper, we analyse the effect of negative words on news consumption. Specifically, we hypothesize that the presence of negative words in a headline will increase the click-through rate (CTR) for that headline. Table 1 shows the design table. Using a text mining framework, we extract negative words and estimate the effect on CTR using a multilevel regression (see Methods). We provide empirical evidence from large-scale RCTs in the field (N = 22,743). Overall, our data contain over 105,000 different variations of news headlines from Upworthy, which have generated ~5.7 million clicks and more than 370 million impressions.

In addition to examining the effect of negative words as our primary analysis, we further conduct a secondary analysis examining the effect of high and low-arousal negative words. Negative sentiment consists of many discrete negative emotions. Previous work has proposed that certain discrete categories of negative emotions may be especially attention-grabbing[58]. For example, high-arousal negative emotions such as anger or fear have been found to efficiently attract attention and be quickly recognizable in facial expressions and body language[31,59,60]. This may be because of the social and informational value that high-arousal emotions such as anger and fear hold—both could alert others in one's group to threats, and paying preferential attention and recognition to these emotions could help the group survive[27,32]. This may also be why in the current age, people are more likely to share and engage with online content that is embedding anger,

**Table 2 | Example experiments (RCTs) performed by Upworthy**

| # | Headline variation | CTR (%) |
|---|---|---|
| 1 | If The Numbers 4 And 20 Mean Something To You, You're Gonna Want To Hear This ***Shit*** | 0.94 |
| 2 | What He Has To Say About Pot Is Going To Make Both Sides ***Angry***, But Here He Goes | 0.79 |
| 3 | Lots Of Things In Live Have Both A **Benefit** And A ***Harm***, So Why Do We Only ***Obsess*** About This One? | 0.60 |
| 4 | He Explains Why The Question 'What Are You Smoking' Is Actually **Kind** of **Important**. | 0.58 |
| 1 | IMAGINE: You're ***Raped*** At Your Job And Your Boss Intentionally Tries To Shut You Up | 0.92 |
| 2 | 12 Minutes. If You **Support** Our Troops, Sacrifice at Least That Much To Them. | 0.21 |
| 1 | Spoofers Set Up A ***Fake*** Agency To Show How ***Ridiculous*** Some People Are When It Comes To Immigrants. | 0.65 |
| 2 | Something's Been ***Missing*** From Our **Favourite** Superhero Stories, And It Makes Reality Seem Kinda ***Silly*** | 0.56 |
| 3 | Some Comic Book **Lovers** Might Need To Check Their Politics When They See What These Guys Have In Mind | 0.53 |
| 4 | A New Agency Wants To Get Rid Of All Our **Favorite** Superheroes. I **Laughed** When I Saw Why. | 0.41 |
| 1 | I Knew Which One She's Pick, But It Still ***Crushed*** Me | 1.10 |
| 2 | First She Points To The **Pretty** Child. Then To The ***Ugly*** Child. Then My Heart Breaks. | 0.85 |
| 3 | 1 Little Girl, 5 Cartoons, And 1 ***Heartbreaking*** Answer | 0.83 |
| 4 | What She Says About These Cartoons Says Something Incredibly ***Troubling*** About The World We Live In | 0.66 |

Shown are four experiments, each with different headline variations subject to testing. Columns report the CTR and the positive (in **bold**) and negative words (in ***bold italics***) as classified by the LIWC dictionary.

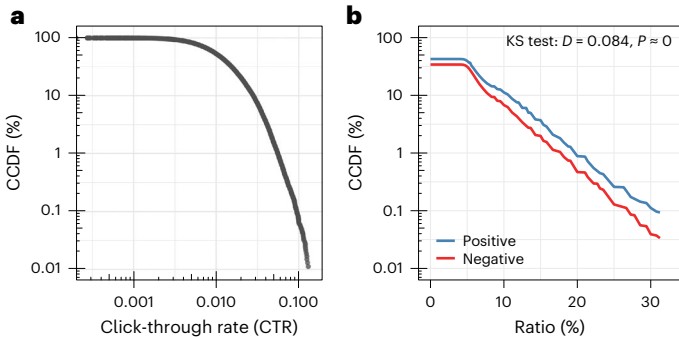

**Fig. 1 | Complementary cumulative distribution function (CCDF). a**, CCDF comparing CTRs across all headline variations ($N$ = 53,669). **b**, CCDF comparing the distribution of the ratio of positive and negative words across all headline variations ($N$ = 53,669). Positive words are more prevalent than negative words. A KS test shows that this difference is statistically significant ($P < 0.001$, two-tailed). The $y$ axes of both plots are on a logarithmic scale.

fear or sadness[21,41,61,62]. Therefore, we examine the effects of words related to anger and fear (as high-arousal negative emotions), as well as sadness (as a low-arousal negative emotion). We also examine the effects of words related to joy (positive emotion), which we predict will be associated with lower CTRs.

## Results

The following analyses are based on a reserved portion of the data (the 'confirmatory sample'), which was only made available after acceptance of a Stage 1 Registered Report. All pilot analyses (reported in the Stage 1 paper) were conducted on a subset of the total data and have no overlap with the analyses for Stage 2. When reporting estimates, we abbreviate standard errors as 'SE' and 99% confidence intervals as 'CI'.

### RCTs comparing news consumption

Our dataset contains a total of $N$ = 22,743 RCTs. These consist of ~105,000 different variations of news headlines from Upworthy.com that generated ~5.7 million clicks across more than 370 million overall impressions. After applying the pre-registered filtering procedure (see Methods), we obtained 12,448 RCTs. Each RCT compares different variations of news headlines that all belong to the same news story. For example, the headline "WOW: Supreme Court Have Made Millions Of Us Very, Very Happy" and "We'll Look Back At This In 10 Years Time And Be Embarrassed As Hell It Even Existed" are different headlines used for the same story about the repeal of Proposition 8 in California. An average of 4.31 headline variations (median of 4) are tested in each RCT. The headline variations are then compared with respect to the generated CTR, defined as the ratio of clicks per impression (see Table 2 for examples). Overall, the 12,448 RCTs comprise 53,699

different headlines, which received over 205 million impressions and 2,778,124 clicks.

In the experiments, the recorded CTRs range from 0.00% to 14.89%. The average CTR across all experiments is 1.39% and the median click rate is 1.07%. Furthermore, the distribution among CTRs is right-skewed, indicating that only a small proportion of news stories are associated with a high CTR (Fig. 1a). For instance, 99% of headline variations have a CTR below 6%. The results lay the groundwork for identifying the drivers of high levels of news consumption (additional descriptive statistics are in Supplementary Table 1 and Fig. 1).

There are considerable differences between positive and negative language in news headlines (Fig. 1b). We find that positive words are more prevalent than negative words (Kolmogorov-Smirnov (KS) test: $D$ = 0.574, $P < 0.001$, two-tailed). Overall, 2.83% of all words in news headlines are categorized as positive words, whereas 2.62% of all words are categorized as negative words. In our sample, the most common positive words are 'love' ($n$ = 980), 'pretty' ($n$ = 746) and 'beautiful' ($n$ = 645), and the most common negative words are 'wrong' ($n$ = 728), 'bad' ($n$ = 588) and 'awful' ($n$ = 363). Ninety percent (91.61%) of the news stories in our sample contain a headline with at least one positive or negative word (that is, 11,403 out of a possible 12,448) and 63.55% of headlines contain at least one word from our dictionaries (that is, 34,124 out of a possible 53,699). Further statistics with word frequencies are in Supplementary Table 2.

### Effect of negative language on news consumption

Randomized controlled experiments were used to estimate the effect of positive and negative words on news consumption, that is, the CTR. We employed a multilevel binomial regression that accommodates a random effects specification to capture heterogeneity among news stories (for details, see Methods; coefficient estimates are in Supplementary Table 3).

Positive and negative language in news headlines are both important determinants of CTRs (Fig. 2a–c). Consistent with the 'negativity bias hypothesis', the effect for negative words is positive ($\beta$ = 0.015, SE = 0.001, $z$ = 17.423, $P < 0.001$, 99% CI = (0.013, 0.018)), suggesting that a larger proportion of negative words in the headline increases the propensity of users to access a news story. A one standard deviation larger proportion of negative words increases the odds of a user clicking the headline by 1.5%. For a headline of average length (14.965 words), this implies that for each negative word, the CTR increases by 2.3%. In contrast, the coefficient for positive words is negative ($\beta$ = −0.008, SE = 0.001, $z$ = −9.238, $P < 0.001$, 99% CI = (−0.010, −0.006)), implying that a larger proportion of positive words results in fewer clicks.

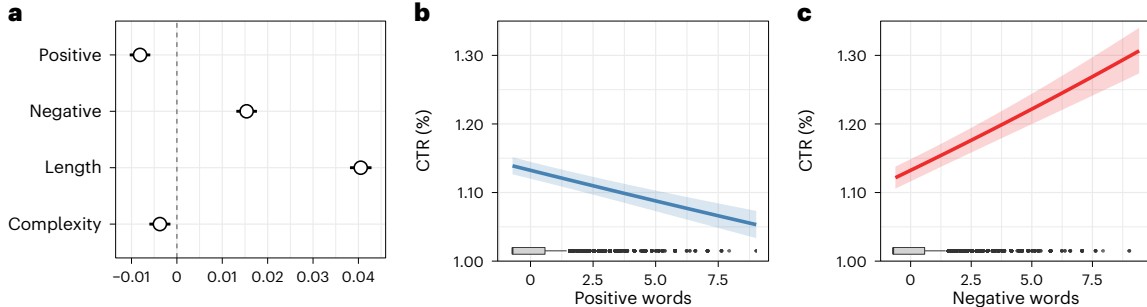

**Fig. 2 | The effect of positive and negative words in news headlines on the CTR.** Headlines (*N* = 53,669) were examined over 12,448 RCTs. **a**, Estimated standardized coefficients (circles) with 99% confidence intervals (error bars) for positive and negative words and for further controls. The variable 'PlatformAge' is included in the model during estimation but not shown for better readability. Full estimation results are in Supplementary Table 3. **b**,**c**, Predicted marginal effects on the CTR (lines). The error bands (shaded area) correspond to 99% confidence intervals. Boxplots show the distribution of the variables in our sample (centre line gives the median, box limits are upper and lower quartiles, whiskers denote minimum/maximum, points are outliers defined as being beyond 1.5× the interquartile range).

---

**Table 3 | Results of the regression model with varying slopes for the proportion of positive and negative words**

| | Coefficient | Lower CI | Upper CI | *P* value |
|---|---|---|---|---|
| Positive | −0.017 | −0.023 | −0.010 | <0.001 |
| Negative | 0.018 | 0.011 | 0.025 | <0.001 |
| Length | 0.044 | 0.041 | 0.047 | <0.001 |
| Complexity | −0.002 | −0.005 | 0.001 | 0.1 |
| PlatformAge | −0.311 | −0.325 | −0.296 | <0.001 |
| (Intercept) | −4.491 | −4.506 | −4.476 | <0.001 |
| Observations: 53,699 | | | | |

Experiment-specific intercepts and slopes for the sentiment variables (that is, random effects) are also included. Reported are standardized coefficient estimates and 99% CIs. *P* values are calculated using two-sided *z*-tests. *N* =53,699 headlines examined over 12,448 RCTs.

For each standard deviation increase in the proportion of positive words per headline, the likelihood of a click decreases by 0.8%. Put differently, for each positive word in a headline of average length, the CTR decreases by 1.0%.

The estimated effects hold when adjusting for length and text complexity. A longer news headline increases the CTR ($\beta$ = 0.040, SE = 0.001, $z$ = 43.945, $P$ < 0.001, 99% CI = (0.038, 0.043)). The CTR is decreased by a higher complexity score ($\beta$ = 0.004, SE = 0.001, $z$ = −4.163, $P$ < 0.001, 99% CI = (−0.006, −0.001)), albeit to a smaller extent. This finding implies that lengthier and less complex formulations are appealing to users and lead to higher levels of news consumption. The control for platform age is negative ($\beta$ = −0.309, SE = 0.005, $z$ = −56.917, $P$ < 0.001, 99% CI = (−0.323, −0.295)). Hence, stories published later in Upworthy's career had lower CTRs than stories published at the beginning of Upworthy's career, implying that Upworthy headlines were most successful when its editorial practices were novel to online users.

### Regression analysis with varying slopes

Following our pre-registration, we further report results from a regression analysis with random effects and additional varying slopes in the sentiment variables (Table 3). As such, the receptivity to language is no longer assumed to be equal across all experiments but is allowed to vary. Again, the coefficients are negative for positive words and positive for negative words. This thus implies that positive language decreases the clickability of news headlines, while negative language increases it. Furthermore, this is consistent with the analysis based on a random effects model without varying slopes. Importantly, the results from both the varying-slopes model and our main model align with our pre-registered hypotheses, providing converging evidence of a negativity bias in news consumption.

Altogether, we find that a higher share of negative language in news headlines increases the CTR, whereas a higher share of positive language decreases the CTR. It is important to note that headlines belong to the 'same' news story and, therefore, phrasing news, regardless of its story, in a negative language increases the rate of clicking on a headline.

### Robustness checks

The robustness of our preliminary analysis was confirmed by a series of further checks (see Supplement D). First, we repeated the analysis with alternative sentiment dictionaries as an additional validation. Specifically, we examined both the NRC dictionary[63,64] and the SentiStrength dictionary[65]. We found that the coefficient estimates were in good agreement, contributing to the robustness of our results (Supplementary Table 4 and Fig. 2). Second, we repeated the above main analysis with an alternative approach for negation handling (that is, a different neighbourhood for inverting the polarity of words). This approach led to qualitatively identical results (Supplementary Table 5 and Fig. 3). Third, we repeated the analyses above using alternate text complexity measures. We found that there is a robust effect for both positive and negative words (Supplementary Table 6 and Fig. 4). Fourth, we controlled for quadratic effects. We still observed a dominant effect of negative language (Supplementary Table 7 and Fig. 5). Fifth, we repeated the analyses above but removed headlines where both positive and negative words were simultaneously present. As such, we ended up with all headlines that exclusively included either positive or negative words. We found that headlines with negative words continued to be more likely to be clicked on than headlines with positive words (Supplementary Table 9). Sixth, we repeated the same analyses as above, but removed all image RCTs where the teaser images were varied. This approach led to nearly identical results (Supplementary Table 9). Seventh, we computed a single sentiment score, which is given by the net difference between the proportion of positive words and the proportion of negative words. As expected, negative sentiment increased CTR (Supplementary Table 10). All aforementioned robustness checks were included in the Stage 1 Analysis Plan. During Stage 2 revisions, one reviewer asked us to check whether the results were robust when the CTR was log-transformed because of the positive skew of the data. While this analysis was not registered in our Stage 1 paper, we found that negative sentiment continued to increase CTRs (Supplementary Table 11).

We investigated moralized language as a possible moderator of positive and negative language in driving the CTR. Previously,

moralized language was identified as an important driver of the diffusion of social media content[39]. We extended the regression models from our main analysis with interaction terms between the proportion of moral words per headline and the variables for the proportion of positive and negative words. In addition, we included the proportion of moral words per headline as a regressor to estimate its direct effect. We found a negative and statistically significant direct effect of moralized language on CTR ($\beta = -0.024$, SE = 0.001, $z = -17.067$, $P < 0.001$, 99% CI = (−0.028, −0.020)), and negative and statistically significant effects for the interactions between the proportion of moral words and the proportion of positive ($\beta = -0.006$, SE = 0.001, $z = -5.321$, $P < 0.001$, 99% CI = (−0.010, −0.003)) and negative words ($\beta = -0.007$, SE = 0.001, $z = -7.048$, $P < 0.001$, 99% CI = (−0.010, −0.005)). Full model results are shown in Supplementary Table 12 and Fig. 6. The direct negative effect of moral words suggests that headlines that contained moral words were less likely to be clicked on. The interactions suggest that positive words have a more negative effect and negative words have a less positive effect when moral words are also present in the headline. The results thus point towards a direct and a moderating role of moralized language. Yet even when controlling for moralized language, the direct effect of negative language was still present and continues to support the 'negativity bias hypothesis'.

### Negativity effect across different news topics

We examined the effect of negative language across various news topics. The rationale is that news stories in our data comprise various topics for which the effect of negativity on the CTR could potentially differ. To this end, we applied topic modelling as in earlier research (for example, refs. [22,66]). Topic modelling infers a categorization of large-scale text data through a bottom-up procedure, thereby grouping similar content into topics. To obtain topic labels for each headline, we employed the topic model from our pre-registration phase (see Stage 1 of the Registered Report) that groups headlines into 7 topics: 'Entertainment', 'Government & Economy', 'LGBT', 'Parenting & School', 'People' and 'Women Rights & Feminism'. We applied this topic model to infer topic labels for each headline in our dataset. Characteristic words for each topic are shown in Supplementary Table 13 and representative headlines for each topic are shown in Supplementary Table 14.

The topic labels were validated to check whether they provide meaningful representations. In a user study, participants were shown headlines from each topic and were asked to select which topic the headline best fit into (topic intrusion test). Participants ($k = 10$) identified the topic from the correct headline in 51.1% of the cases. For comparison, a random guess would lead to an accuracy of 25%, implying that participants are roughly twice as good. Further, this improvement over the random guess was statistically significant ($\chi^2 = 249.61$, $P < 0.001$). Details are provided in Supplementary Table 15.

We found significant differences in the baseline CTR among topics. For this, we estimated a model where we additionally control for different topics via dummy variables, thus capturing the heterogeneity in how different topics generate clicks. Keeping everything else equal, we find that news generated more clicks when covering stories related to 'Entertainment', 'LGBT' and 'People'. In contrast, news related to 'Government & Economy' have a lower clickability (Supplementary Table 16).

We then controlled for how the effect of negative language might vary across different topics. Here we found that the variables of interest (that is, the proportion of positive and negative words) significantly interact with different topics. For example, headlines relating to 'Government & Economy', 'LGBT', 'Parenting & School' and 'People' received more clicks when they contained a large share of negative words. We also found that headlines relating to 'LGBT', 'Life', 'Parenting & School' and 'People' received fewer clicks when they contained a

large share of positive words. Overall, we found that negative language still has a statistically significant positive effect on the CTR (Supplementary Table 17). In sum, these results are consistent with the main analysis.

### Extension to discrete emotions

We conducted secondary analyses examining the effects of discrete emotional words on the CTR (summary statistics are given in Supplementary Fig. 7 and Table 18). Previous work has suggested that certain discrete emotions such as anger[38,41] may be particularly prevalent in online news. Furthermore, discrete emotions were found to be important determinants of various forms of user interactions (for example, sharing[36–41]), thus motivating the idea that discrete emotions may also play a role in news consumption.

We report findings from four emotions (anger, fear, joy, sadness) for which we found statistically significant positive correlations between the human judgments of emotions and the dictionary scores (Methods). We observed a statistically significant and positive coefficient for sadness ($\beta = 0.006$, SE = 0.001, $z = 5.295$, $P < 0.001$, 99% CI = (0.003, 0.009)) and a statistically significant negative effect for joy ($\beta = -0.009$, SE = 0.001, $z = -7.664$, $P < 0.001$, 99% CI = (−0.012, −0.006)) and fear ($\beta = -0.007$, SE = 0.001, $z = -5.919$, $P < 0.001$, 99% CI = (−0.009, −0.004)). A one standard deviation increase in sadness increases the odds of a user clicking the headline by 0.7%, while a one standard deviation increase in joy or fear decreases the odds of a user clicking on a headline by 0.9% and 0.7%, respectively. The coefficient estimate for anger ($\beta = 0.000$, SE = 0.001, $z = -0.431$, $P = 0.666$, 99% CI = (−0.003, 0.002)) was not statistically significant at common statistical significance thresholds.

We performed an equivalence test[67] to see whether the relationship between the emotion score for anger and the CTR can be dismissed as null. The equivalence test involves defining a threshold for the smallest meaningful effect (here: −0.001 to +0.001) and determining whether the effect of interest falls within that threshold. We find that the equivalence test for anger was undecided ($P = 0.399$, 99% CI = (−0.003, 0.002)), suggesting that the results are inconclusive with respect to a null effect. Our results for joy and sadness were consistent with our pre-registered hypotheses, while our results for anger and fear did not align with our hypotheses. Consistent with our previous findings, we observed that the CTR increases as the text length increases ($\beta = 0.037$, SE = 0.001, $z = 33.206$, $P < 0.001$, 99% CI = (0.034, 0.040)) and decreases as the complexity score increases ($\beta = -0.004$, SE = 0.001, $z = -3.410$, $P < 0.001$, 99% CI = (−0.007, −0.001)). Again, the CTR was lower for headlines at the end of Upworthy's career ($\beta = -0.312$, SE = 0.006, $z = -55.029$, $P < 0.001$, 99% CI = (−0.327, −0.298)). Full results are given in Supplementary Table 19.

The above findings are supported by additional checks. First, we controlled for different topics in our regression model, utilizing the previous categorization via topic modelling. When including topic dummies, we still found statistically significant positive effects for sadness and significant negative effects for fear and joy; the coefficient for anger was not statistically significant (Supplementary Table 20). These results are thus consistent with the main analysis. For thoroughness, we also conducted exploratory analyses using the four other basic emotions from the NRC emotion dictionary for which the correlation with human judgments was considerably lower in our validation study (Supplementary Tables 21 and 22). Here we observed a statistically significant negative effect on the CTR for anticipation. Furthermore, we studied the effects of 24 emotional dyads from Plutchik's model[68], these dyads being complex emotions composed of two basic emotions[69]. Consistent with findings from previous literature[21,43,61], we found that several dyads such as outrage and disapproval are associated with higher CTRs. The full results of our exploratory analyses on basic and higher-order emotions can be found in Supplementary Figs. 8–11 and Tables 23–25.

## Discussion

To examine the causal impact of emotional language on news consumption, we analysed a large dataset[53] of more than 105,000 headlines that encompassed more than 370 million impressions of news stories from Upworthy.com. Consistent with our pre-registration, we find supporting evidence for a negativity bias hypothesis: news headlines containing negative language are significantly more likely to be clicked on, even after adjusting for the corresponding content of the news story. For a headline of average length (~15 words), the presence of a single negative word increased the CTR by 2.3%. In contrast, we find that news headlines containing positive language are significantly less likely to be clicked on. For a headline of average length, the presence of positive words in a news headline significantly decreases the likelihood of a headline being clicked on, by around 1.0%. The effects of positive and negative words are robust across different sentiment dictionaries and after adjusting for length, text complexity and platform age. While the observed effect sizes were noticeably smaller than in studies analysing sharing behaviour[11], increases/decreases of around 1–2% have been found to be meaningful differences when studying negativity bias in news consumption[45]. Notably, we uncovered a negativity bias in the data even though Upworthy branded itself as a 'positive news outlet'[53]. Hence, while Upworthy readers may have chosen this outlet for its positive spin on the news, negative language increased news consumption, while positive language decreased it.

We compared the effect of negative words across different news topics. Our analyses revealed that the positive effect of negative words on consumption rates was strongest for news stories pertaining to 'Government & Economy'. These results suggest that consistent with previous work, individuals are especially likely to consume political and economic news when it is negative[46–50,52]. Hence, people may be (perhaps unintentionally) selectively exposing themselves to divisive political news, which ultimately may contribute to political polarization and intergroup conflict[45,61,70].

We further extended our analyses to examine the effect of discrete emotional language on news consumption. Consistent with our pre-registered hypotheses, basic emotions such as 'sadness' increased the likelihood that a news headline was clicked on, while 'joy' decreased the likelihood that a news headline was clicked on. Interestingly, we did not see a statistically significant effect for words related to 'anger', which previous research has suggested to play a role in online diffusion[22,38]. Furthermore, the effect of 'fear' was significantly related to article consumption, but in the opposite direction of our pre-registered hypotheses. Interestingly, much work thus far has found that high valence emotions (such as 'fear' and 'anger') are related to online sharing behaviour[34,35,37,44]. Here, however, we measured online news consumption, a private behaviour, which may account for differences from previous findings. Consumption behaviour must be measured unobtrusively, so that it captures all content that individuals want to attend to, instead of what they want others to know they attended to. Sharing behaviour is public, curated and influenced by a myriad of social factors (for example, signalling group identity, maintaining reputation)[31,61,71]. The distinction between engaging with content online and consuming content online is important, as social considerations play a role in the decision to share content, but not necessarily in consuming content[61]. Generally, people share only a fraction of the content they consume online, implying that engagement may be driven by different emotions and goals. These different motivations may make 'fear' and 'anger' more influential in the decision to share, compared with the decision to consume. Here, the Upworthy Research Archive gives us a large-scale opportunity to examine people's personal preference for news they attend to as opposed to news they want to share with others. Future research should investigate the differences between sharing behaviour and consumption behaviour (for example, considering personality[72]).

Upworthy's use of large-scale RCT testing allows social scientists the opportunity to analyse real-world behaviour from millions of users in an experimental setting and thereby make causal inferences. Furthermore, Upworthy was extremely popular and continues to have a lasting impact on editorial practices through the media landscape[23,56,57]. We found that headlines received fewer clicks when they were published later in Upworthy's career. This might suggest that Upworthy's 'click-bait' editorial practices were most effective when they were novel to readers and that the effectiveness decayed over time. This should make our data representative of news consumption across many contemporary online media sites. The results of our study demonstrate a robust and causal negativity bias in news consumption from a massive dataset from the field.

As with other research, ours is not free from limitations. Upworthy differs from more traditional news sources due to its use of 'click-bait' headlines. Nevertheless, we think that analysing user behaviour at Upworthy is important due to its large readership and influence on the media landscape. Moreover, it is important to note that we can only draw conclusions at the level of news stories but not at the level of individuals. Along similar lines, we are limited in the extent to which we can infer the internal state of a perceiver on the basis of the language they write, consume or share[73]. In general, using online data to infer individual differences in subjective feelings is challenging. However, our analysis does not attempt to infer subject feelings; rather, we quantify how the presence of certain words is linked to concrete behaviour. Hence, readers' preference for headlines containing negative words does not imply that users felt more negatively while reading said headlines. Our results show that negative words increase consumption rates, but make no claims regarding the subjective experience of readers.

Another potential limitation of these analyses is our use of discrete emotions via the NRC emotion lexicon. Our choice of the NRC emotion lexicon was due to two reasons: first, the NRC emotion lexicon is a prominent and comprehensive choice for examining discrete emotions in text[64] and, second, it captures various emotions at a granular level. In this regard, previous work has posited that specific emotions, such as 'anger', 'outrage' or 'disapproval', are prevalent in online news[21,43,58,74]. Motivated by this, we felt it necessary to include an analysis of discrete emotions for comparability and richness. Nevertheless, following a psychological constructionist perspective, models of emotion involving a 2 × 2 dimensional space for valence and arousal have been proposed[71,75]; yet the availability of corresponding dictionaries for such emotion models are limited, hence we opted for the NRC emotion lexicon.

Understanding the biases that influence people's consumption of online content is critical, especially as misinformation, fake news and conspiracy theories proliferate online. Even publishers marketed as 'good news websites' are benefiting from negativity, demonstrating the need for a nuanced understanding of news consumption. Knowing what features of news make articles interesting to people is a necessary first step for this purpose and will enable us to increase online literacy and to develop transparent online news practices.

## Methods

### Ethics information

The research complies with all relevant ethical regulations. Ethics approval (2020-N-151) for the main analysis was obtained from the Institutional Review Board (IRB) at ETH Zurich. For the user validation, ethics approval (IRB-FY2021-5555) was obtained from the IRB at New York University. Participants in the validation study were recruited from the subject pool of the Department of Psychology at New York University in exchange for 0.5 h of research credit for varying psychology courses. Participants provided informed consent for the user validation studies. New York University did not require IRB approval for the main analysis, as it is not classified as a human subjects research.

## Large-scale field experiments

In this research, we build upon data from the Upworthy Research Archive[53]. The data have been made available through an agreement between Cornell University and Upworthy. We have access to this dataset upon the condition of following the procedure for a Registered Report. In Stage 1, we had access only to a subset of the dataset (that is, the 'exploratory sample'), on the basis of which we conducted the preliminary analysis for pre-registering hypotheses. In Stage 2 (this paper), we had access to a separate subset of the data (that is, the 'confirmatory sample') on the basis of which we tested the pre-registered hypotheses. Here, our analysis was based on data from $N = 22{,}743$ experiments (RCTs) collected on Upworthy between 24 January 2013 and 14 April 2015.

Each RCT corresponds to one news story, in which different headlines for the same news story were compared. Formally, for each headline variation $j$ in an RCT $i$ ($i = 1, \dots, N$), the following statistics were recorded: (1) the number of impressions, that is, the number of users to whom the headline variation was shown (impressions$_{ij}$) and (2) the number of clicks a headline variation generated (clicks$_{ij}$). The CTR was then computed as $\text{CTR}_{ij} = \frac{\text{clicks}_{ij}}{\text{impressions}_{ij}}$. The experiments were conducted separately (that is, only a single experiment was conducted at the same time for the entire website) so each test can be analysed as independent of all other tests[53]. Examples of news headlines in the experiments are presented in Table 2. The Upworthy Research Archive contains data aggregated at the headline level and, thus, does not provide individual-level data for users.

The data were subjected to the following filtering. First, all experiments solely consisting of a single headline variation were discarded. Single headline variations exist because Upworthy conducted RCTs on features of their articles other than headlines, predominantly teaser images. In many RCTs where teaser images were varied, headlines were not varied at all (image data were not made available to researchers by the Upworthy Research Archive, so we were unable to incorporate image RCTs into our analyses although we validated our findings as part of the robustness checks). Second, some experiments contained multiple treatment arms with identical headlines, which were merged into one representative treatment by summing their clicks and impressions. These occurred when images 'and' headlines were involved in RCTs for the same story. This is relatively rare in the dataset, but for robustness checks regarding image RCTs, see Supplementary Table 9.

The analysis in the current Registered Report Stage 2 is based on the confirmatory sample of the dataset[53], which was made available to us only after pre-registration was conditionally accepted. In the previous pre-registration stage, we presented the results of a preliminary analysis based on a smaller, exploratory sample (see Registered Report Stage 1). Both were processed using identical methodology. The pilot sample for our preliminary analysis comprised 4,873 experiments, involving 22,666 different headlines before filtering and 11,109 headlines after filtering, which corresponds to 4.27 headlines on average per experiment. On average, there were approximately 16,670 participants in each RCT. Additional summary statistics are given in Supplementary Table 1.

## Design

We present a design table summarizing our methods in Table 1.

## Sampling plan

Given our opportunity to secure an extremely large sample where the $N$ was predetermined, we chose to run a simulation before pre-registration to estimate the level of power we would achieve for observing an effect size represented by a regression coefficient of 0.01 (that is, a 1% effect on the odds of clicks from a standard deviation increase in negative words). This effect size is slightly more conservative than estimates of effect sizes from pilot studies (see Stage 1

of the Registered Report) and is derived from theory[76]. The size of the confirmatory Upworthy data archive is $N = 22{,}743$ RCTs, with between 3 and 12 headlines per RCT. This thus corresponds to a total sample of between 68,229 and 227,430 headlines. Because we were not aware of the exact size during pilot testing, we generated datasets through a bootstrapping procedure that sampled $N = 22{,}743$ RCTs with replacement from our pilot sample of tests. We simulated 1,000 such datasets and for each dataset we generated 'clicks' using the estimated parameters from the pilot data. Finally, each dataset was analysed using the model as described. This procedure was repeated for both models (varying intercepts, and a combination of varying intercepts and varying slopes). We found that under the assumptions of effect size, covariance matrix and data generating process from our pilot sample, we will have greater than 99% power to detect an effect size of 0.01 in the final sample for both models.

## Analysis plan

**Text mining framework.** Text mining was used to extract emotional words from news headlines. To prepare the data for the text mining procedure, we applied standard preprocessing to the headlines. Specifically, the running text was converted into lower-case and tokenized, and special characters (that is, punctuations and hashtags) were removed. We then applied a dictionary-based approach analogous to those of earlier research[22,39-41].

We performed sentiment analysis on the basis of the Linguistic Inquiry and Word Count (LIWC)[77]. The LIWC contains word lists classifying words according to both a positive ($n = 620$ words, for example 'love' and 'pretty') and negative sentiment ($n = 744$ words, for example 'wrong' and 'bad'). A list of the most frequent positive and negative words in our dataset is given in Supplementary Table 2.

Formally, sentiment analysis was based on single words (that is, unigrams) due to the short length of the headlines (mean length: 14.965 words). We counted the number of positive words ($n_{\text{positive}}$) and the number of negative words ($n_{\text{negative}}$) in each headline. A word was considered 'positive' if it is in the dictionary of positive words (and vice versa, for 'negative' words). We then normalized the frequency by the length of the headline, that is, the total number of words in the headline ($n_{\text{total}}$). This yielded the two separate scores

$$\text{Positive}_{ij} = \frac{n_{\text{positive}}}{n_{\text{total}}} \text{ and } \text{Negative}_{ij} = \frac{n_{\text{negative}}}{n_{\text{total}}}$$

for headline $j$ in experiment $i$. As such, the corresponding scores for each headline represent percentages. For example, if a headline has 10 words out of which one is classified as 'positive' and none as 'negative,' the scores are $\text{Positive}_{ij} = 10\%$ and $\text{Negative}_{ij} = 0\%$. If a headline has 10 words and contains one 'positive' and one 'negative' word, the scores are $\text{Positive}_{ij} = 10\%$ and $\text{Negative}_{ij} = 10\%$. A headline may contain both positive and negative words, so both variables were later included in the model.

Negation words (for example, 'not,' 'no') can invert the meaning of statements and thus the corresponding sentiment. We performed negation handling as follows. First, the text was scanned for negation terms using a predefined list, and then all positive (or negative) words in the neighbourhood were counted as belonging to the opposite word list, that is, they were counted as negative (or positive) words. In our analysis, the neighbourhood (that is, the so-called negation scope) was set to 3 words after the negation. As a result, a phrase such as 'not happy' was coded as negative rather than positive. Here we used the implementation from the sentimentr package (details at https://cran.r-project.org/web/packages/sentimentr/readme/README.html).

Using the above dictionary approach, our objective was to quantify the presence of positive and negative words. As such, we did not attempt to infer the internal state of a perceiver on the basis of the language they write, consume or share[73]. Specifically, readers'

preference for headlines containing negative words does not imply that users 'felt' more negatively while reading said headlines. In contrast, we quantified how the presence of certain words is linked to concrete behaviour. Following this, our pre-registered hypotheses test whether negative words increase consumption rates (Table 1).

We validated the dictionary approach in the context of our corpus on the basis of a pilot study[78]. Here we used the positive and negative word lists from LIWC[77] and performed negation handling as described above. Perceived judgments of positivity and negativity in headlines correlate with the number of negative and/or positive words each headline contains. Specifically, we correlated the mean of the 8 human judges' scores for a headline with NRC sentiment rating for that headline. We found a moderate but significant positive correlation ($r_s = 0.303$, $P < 0.001$). These findings validate that our dictionary approach captures significant variation in the perception of emotions in headlines from perceivers. More details are available in Supplementary Tables 21 and 22.

Two additional text statistics were computed: first, we determined the length of the news headline as given by the number of words. Second, we calculated a text complexity score using the Gunning Fog index[79]. This index estimates the years of formal education necessary for a person to understand a text upon reading it for the first time: $0.4 \times (\text{ASL} + 100 \times n_{\text{wsy}\geq3}/n_w)$, where ASL is the average sentence length (number of words), $n_w$ is the total number of words and $n_{\text{wsy}\geq3}$ is the number of words with three syllables or more. A higher value thus indicates greater complexity. Both headline length and the complexity score were used as control variables in the statistical models. Results based on alternative text complexity scores are reported as part of the robustness checks.

The above text mining pipeline was implemented in R v4.0.2 using the packages quanteda (v2.0.1) and sentimentr (v2.7.1) for text mining.

**Empirical model.** We estimated the effect of emotions on online news consumption using a multilevel binomial regression. Specifically, we expected that negative language in a headline affects the probability of users clicking on a news story to access its content. To test our hypothesis, we specified a series of regression models where the dependent variable is given by the CTR.

We modelled news consumption as follows: $i = 1, \ldots, N$ refers to the different experiments in which different headline variations for news stories are compared through an RCT; $\text{clicks}_{ij}$ denote the number of clicks from headline variation $j$ belonging to news story $i$. Analogously, $\text{impressions}_{ij}$ refer to the corresponding number of impressions. We followed previous approaches[80] and modelled the number of clicks to follow a binomial distribution as

$$\text{clicks}_{ij} \sim \text{Binomial}(\text{impressions}_{ij}, \theta_{ij})$$

where $0 \leq \theta_{ij} \leq 1$ is the probability of a user clicking on a headline in a single Bernoulli trial and where $\theta_{ij}$ corresponds to the CTR of headline variation $j$ from news story $i$.

We estimated the effect of positive and negative words on the CTR $\theta_{ij}$ and captured between-experiment heterogeneity through a multilevel structure. We further controlled for other characteristics across headline variations, namely length, text complexity and the relative age of a headline (based on the age of the platform). The regression model is then given by

$$\text{logit}(\theta_{ij}) = \alpha + \alpha_i + \beta_1 \text{Positive}_{ij} + \beta_2 \text{Negative}_{ij}$$
$$+ \gamma_1 \text{Length}_{ij} + \gamma_2 \text{Complexity}_{ij} + \gamma_3 \text{PlatformAge}_{ij}$$

where $\alpha$ is the global intercept and $\alpha_i$ is an experiment-specific intercept (that is, random effect). Both $\alpha$ and $\alpha_i$ are assumed to be independent and identically normally distributed with a mean of zero. The latter

captures heterogeneity at the experiment level; that is, some news stories might be more interesting than others. In addition, we controlled for the length ($\text{Length}_{ij}$) and complexity ($\text{Complexity}_{ij}$) of the text in the news headline, as well as the relative age of the current experiment with regard to the platform ($\text{PlatformAge}_{ij}$). The latter denotes the number of days of the current experiment since the first experiment on Upworthy.com in 2012 and thus allowed us to control for potential learning effects as well as changes in editorial practices over time. The coefficient $\beta_2$ is our main variable of interest: it quantifies the effect of negative words on the CTR.

In the above analysis, all variables were z-standardized for better comparability. That is, before estimation, we subtracted the sample mean and divided the difference by the standard deviation. Because of this, the regression coefficients $\beta_1$ and $\beta_2$ quantify changes in the dependent variable in standard deviations. This allowed us to compare the relative effect sizes across positive and negative words (as well as emotional words later). Due to the logit link, the odds ratio is $100 \times (e^\beta - 1)$, which gives the percentage change in the odds of success as a result of a standard deviation change in the independent variable. In our case, a successful event is indicated by the user clicking the headline.

The above regression builds upon a global coefficient for capturing the effect of language on CTR and, as such, the language reception is assumed to be equal across different RCTs. This is consistent with previous works where a similar global coefficient (without varying slopes) was used[22,34,38,39]. However, there is reason to assume that the receptivity to language might vary across RCTs and thus among news (for example, the receptivity of negative language might be more dominant for political news than for entertainment news, or for certain news topics over others). As such, the variance in the estimated regression coefficients is no longer assumed to be exactly zero across experiments but may vary. To do so, we augmented the above random effects model by an additional varying-slopes specification. Here, a multilevel structure was used that accounts for the different levels due to the experiments $i = 1, \ldots, N$. Specifically, the coefficients $\beta_1$ and $\beta_2$ capturing the effect of positive and negative words on CTR, respectively, were allowed to vary across experiments. Of note, a similar varying-slopes formalization was only used for the main analysis on the basis of positive and negative language, and not for the subsequent extension to emotional words where it is not practical due to the fact that there would be comparatively fewer treatment arms in comparison with the number of varying slopes.

Here we conducted the analysis on the basis of both models, that is, (1) the random effect model and (2) the random effect model with additional varying slopes. If the estimates from both models are in the same direction, this should underscore the overall robustness of the findings. If estimated coefficients from the random effect model and the random effect, varying-slopes model contradict each other, both results are reported but precedence in interpretation is given to the latter due to its more flexible specification.

All models were estimated using the lme4 package (v1.1.23) in R.

**Extension to discrete emotional words.** To provide further insights into how emotional language relates to news consumption, we extended our text mining framework and performed additional secondary analyses. We were specifically interested in the effect of different emotional words (anger, fear, joy and sadness) on the CTR.

Here, our analyses were based on the NRC emotion lexicon due to its widespread use in academia and the scarcity of other comparable dictionaries with emotional words for content analysis[63,64]. The NRC lexicon comprises 181,820 English words that are classified according to the 8 basic emotions of Plutchik's emotion model[67]. Basic emotions are regarded as universally recognized across cultures and on this basis, more complex emotions can be derived[69,81]. The 8 basic emotions computed via the NRC were anger, anticipation, joy, trust, fear, surprise, sadness and disgust.

We calculated scores for basic emotions embedded in news headlines on the basis of the NRC emotion lexicon[63]. We counted the frequency of words in the text that belong to a specific basic emotion in the NRC lexicon (that is, an 8-dimensional vector). A list of the most frequent emotional words in our dataset is given in Supplementary Table 18. Afterwards, we divided the word counts by the total number of dictionary words in the text, so that the vector is normalized to sum to one across the basic emotions. Following this definition, the embedded emotions in a text might be composed of, for instance, 40% 'anger' while the remaining 60% are 'fear'. We omitted headline variations that do not contain any emotional words from the NRC emotion lexicon (since, otherwise, the denominator was not defined). Due to this extra filtering step, we obtained a final sample of 39,897 headlines. We again accounted for negations using the above approach in that the corresponding emotional words are not attributed to the emotion but skipped during the computation (as there is no defined 'opposite' emotion).

As a next step, we validated the NRC emotion lexicon for the context of our study through a user study. Specifically, we correlated the mean of the 8 human judges' scores for a headline with NRC emotion rating for that headline. We found that overall, both mean user judgments on emotions and those from the NRC emotion lexicon are correlated ($r_s$: 0.114, $P < 0.001$). Furthermore, mean user judgements for four basic emotions were significantly correlated, namely anger ($r_s$: 0.22, $P = 0.005$), fear ($r_s$: 0.29, $P < 0.001$), joy ($r_s$: 0.24, $P = 0.002$) and sadness ($r_s$: 0.30, $P < 0.001$). The four other basic emotions from the NRC emotion lexicon showed considerably lower correlation coefficients in the validation study, namely anticipation ($r_s$: −0.07, $P = 0.341$), disgust ($r_s$: 0.01, $P = 0.926$), surprise ($r_s$: −0.06, $P = 0.414$) and trust ($r_s$: 0.12, $P = 0.122$). Because of this, we did not pre-register hypotheses for them.

The multilevel regression was specified, analogous to the model above but with different explanatory variables, that is,

$$\text{logit}(\theta_{ij}) = \alpha + \alpha_i + \beta_1 \text{Anger}_{ij} + \beta_2 \text{Fear}_{ij}$$
$$+ \beta_3 \text{Joy}_{ij} + \beta_4 \text{Sadness}_{ij} + \gamma_1 \text{Length}_{ij}$$
$$+ \gamma_2 \text{Complexity}_{ij} + \gamma_3 \text{PlatformAge}_{ij}$$

where $\alpha$ and $\alpha_i$ represent the global intercept and the random effects, respectively. Specifically, $\alpha$ is again the global intercept and $\alpha_i$ captures the heterogeneity across experiments $i = 1,...,N$. As above, we included the control variables, that is, length, text complexity and platform age. The coefficients $\beta_1,...,\beta_4$ quantify the effect of the emotional words (that is, anger, fear, joy and sadness) on the CTR.

Again, all variables were $z$-standardized for better comparability (that is, we subtracted the sample mean and divided the difference by the standard deviation). As a result, the regression coefficients quantify changes in the dependent variable in standard deviations. This allows us to compare the relative effect sizes across different emotions.

### Protocol registration
The Stage 1 protocol for this Registered Report was accepted in principle on 11 March 2022. The protocol, as accepted by the journal, can be found at https://springernature.figshare.com/articles/journal_contribution/Negativity_drives_online_news_consumption_Registered_Report_Stage_1_Protocol_/19657452.

### Reporting summary
Further information on research design is available in the Nature Portfolio Reporting Summary linked to this article.

### Data availability
The full data from the randomized controlled experiments in the field are available through the Upworthy Research Archive[53] (https://doi.org/10.1038/s41597-021-00934-7). The data used in

the present paper are available at https://osf.io/uscpf/. The LIWC dictionary[77] is available for purchase (https://www.liwc.app/). The NRC emotion lexicon[63] is publicly available for download (https://nrc.canada.ca/en/research-development/products-services/technical-advisory-services/sentiment-emotion-lexicons). In our analysis, we used the built-in version from the sentimentr package.

### Code availability
Code that supports the findings of our study is available at https://osf.io/uscpf/.

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

## Acknowledgements

We thank John Templeton Foundation Grant No. 61378 and a Russell Sage Foundation grant (G-2110-33990) that funded J.J.V.B. The funders had no role in study design, data collection and analysis, decision to publish or preparation of the paper. We thank Upworthy, as well as J. Nathan Matias, K. Munger, M. Aubin Le Quere and C. Ebersole for making the data available. We also thank W. Brady for helpful feedback.

## Author contributions

C.E.R., N.P., K.S., P.P., J.J.V.B. and S.F. conceived and designed the experiments. C.E.R., N.P. and K.S. analysed the data. C.R., N.P., K.S., P.P., J.J.V.B. and S.F. wrote the paper.

## Funding

## Competing interests

The authors declare no competing interests.

## Additional information

**Correspondence and requests for materials** should be addressed to Jay J. Van Bavel or Stefan Feuerriegel.

# Reporting Summary

## Statistics

For all statistical analyses, confirm that the following items are present in the figure legend, table legend, main text, or Methods section.

| n/a | Confirmed | |
|---|---|---|
| ☐ | ☒ | The exact sample size (*n*) for each experimental group/condition, given as a discrete number and unit of measurement |
| ☐ | ☒ | A statement on whether measurements were taken from distinct samples or whether the same sample was measured repeatedly |
| ☐ | ☒ | The statistical test(s) used AND whether they are one- or two-sided *Only common tests should be described solely by name; describe more complex techniques in the Methods section.* |
| ☐ | ☒ | A description of all covariates tested |
| ☐ | ☒ | A description of any assumptions or corrections, such as tests of normality and adjustment for multiple comparisons |
| ☐ | ☒ | A full description of the statistical parameters including central tendency (e.g. means) or other basic estimates (e.g. regression coefficient) AND variation (e.g. standard deviation) or associated estimates of uncertainty (e.g. confidence intervals) |
| ☐ | ☒ | For null hypothesis testing, the test statistic (e.g. *F*, *t*, *r*) with confidence intervals, effect sizes, degrees of freedom and *P* value noted *Give P values as exact values whenever suitable.* |
| ☒ | ☐ | For Bayesian analysis, information on the choice of priors and Markov chain Monte Carlo settings |
| ☒ | ☐ | For hierarchical and complex designs, identification of the appropriate level for tests and full reporting of outcomes |
| ☐ | ☒ | Estimates of effect sizes (e.g. Cohen's *d*, Pearson's *r*), indicating how they were calculated |

*Our web collection on statistics for biologists contains articles on many of the points above.*

## Software and code

Policy information about availability of computer code

| Data collection | The authors did not use any software for data collection, as the Upworthy Research Archive is an existing data set. |
|---|---|
| Data analysis | R version 4.0.2, including packages  quanteda version 2.0.1, sentimentr version 2.7.1, and lme4 version 1.1.23 - all code is available at https://osf.io/uscpf/ |

For manuscripts utilizing custom algorithms or software that are central to the research but not yet described in published literature, software must be made available to editors and reviewers. We strongly encourage code deposition in a community repository (e.g. GitHub). See the Nature Portfolio guidelines for submitting code & software for further information.

## Data

Policy information about availability of data

All manuscripts must include a data availability statement. This statement should provide the following information, where applicable:
- Accession codes, unique identifiers, or web links for publicly available datasets
- A description of any restrictions on data availability
- For clinical datasets or third party data, please ensure that the statement adheres to our policy

Both the exploratory and confirmatory data sets from Upworthy.com can be accessed here: https://osf.io/jd64p/. These data are from a 3rd party, we did not collect them ourselves, and they have been published before in Nature Scientific Data. The ciitation for the published data set is:
Matias, J. N., Munger, K., Le Quere, M. A., & Ebersole, C. (2021). The Upworthy Research Archive, a time series of 32,487 experiments in US media. Scientific Data,

8(1), 1-6. https://doi.org/10.1038/s41597-021-00934-7
Data for validation studies is available here: https://osf.io/uscpf/.
We used LIWC dictionary 2015. LIWC dictionaries are available for purchase at the following link: https://www.liwc.app/.

## Human research participants

Policy information about studies involving human research participants and Sex and Gender in Research.

| | |
|---|---|
| Reporting on sex and gender | Demographic information from validation participants was not collected, so we cannot report on the sex and gender makeup of the validation sample. |
| Population characteristics | Participants were New York University Undergraduate students who participated for .5 hours of course credit. Participants were fluent in English. No other demographic information was collected. |
| Recruitment | Participants were recruited from the New York University Student Subject Pool on SONA, and participated for .5 hours of course credit. Participants were all undergraduates and New York University who spoke fluent English, and no other demographic information was collected. |
| Ethics oversight | Institutional Review Board at New York University, Protocol IRB-FY2021-5555 and Institutional Review Board at ETH Zurich, Protocol 2020-N-151 |

Note that full information on the approval of the study protocol must also be provided in the manuscript.

## Field-specific reporting

Please select the one below that is the best fit for your research. If you are not sure, read the appropriate sections before making your selection.

☐ Life sciences        ☒ Behavioural & social sciences        ☐ Ecological, evolutionary & environmental sciences

For a reference copy of the document with all sections, see nature.com/documents/nr-reporting-summary-flat.pdf

## Behavioural & social sciences study design

All studies must disclose on these points even when the disclosure is negative.

| | |
|---|---|
| Study description | Quantitative Experimental |
| Research sample | The research sample consisted of Upworthy.com Users. All data are aggregated at the headline level to ensure user privacy, so there is no demographic data for individual users included in the data set. Upworthy.com recorded impressions and clicks from people who visited their website. Thus, participants needed to be comfortable navigating the Internet. People who did not have internet access from 2013-2015 could not be included in our sample. We chose to use this sample because the data represented real-world behavior (i.e. clicking) and was experimentally controlled (due to Upworthy's use of RCTs.).  N = 538,272,878 participant assignments.  See Matias, J. N., Munger, K., Le Quere, M. A., & Ebersole, C. (2021). The Upworthy Research Archive, a time series of 32,487 experiments in US media. Scientific Data, 8(1), 1-6. for more details.<br><br>For the validation study, participants were recruited from the New York University Student Subject Pool on SONA, and participated for .5 hours of course credit. Participants were all undergraduates and New York University who spoke fluent English, and no other demographic information was collected. This sample is not representative due to it consisting of university students. We chose to use New York University undergraduates because they are a convenience sample. |
| Sampling strategy | Participants were sampled from people who navigated to Upworthy.com between 2013 and 2015, thus was a convenience sample. Power analysis in stage 1 revealed we  have greater than 99% power to detect an effect size of 0.01 in the final sample. |
| Data collection | Data was collected from Upworthy.com from 2013-2015 by the staff at Upworthy.com. Because our analyses are on archival data, the people who collected the data were blind to our hypotheses. The data were collected passively, such that no experimenter was ever present to a participant, nor was anyone else.<br>When a person navigated to Upworthy.com, they were randomly assigned to see one possible story variation from a series of possible variations. Upworthy.com tested images, headlines, and lede text, but our analyses focus on headlines only. When a person sued a computer mouse or a touch screen to select to read an article, Upworthy.com recorded it as a click. |
| Timing | 1-24-13 to 4-30-15 |
| Data exclusions | Only RCTs where headlines were varied were used. Final sample was N = 11,109 headlines |
| Non-participation | Not Applicable. The data were collected passively and recorded regular web browsing activity. |
| Randomization | Participants were randomly assigned to see one headline from a possible set in a given RCT when they navigated to Upworthy.com |

| Randomization | To achieve randomization, Upworthy used Ruby language (versions 2.9.3 through 2.3.1) and the RandomSample Method. |
|---|---|

# Reporting for specific materials, systems and methods

We require information from authors about some types of materials, experimental systems and methods used in many studies. Here, indicate whether each material, system or method listed is relevant to your study. If you are not sure if a list item applies to your research, read the appropriate section before selecting a response.

## Materials & experimental systems

| n/a | Involved in the study |
|---|---|
| ☒ | Antibodies |
| ☒ | Eukaryotic cell lines |
| ☒ | Palaeontology and archaeology |
| ☒ | Animals and other organisms |
| ☒ | Clinical data |
| ☒ | Dual use research of concern |

## Methods

| n/a | Involved in the study |
|---|---|
| ☒ | ChIP-seq |
| ☒ | Flow cytometry |
| ☒ | MRI-based neuroimaging |

