## [Peer Review File · Nature Human Behaviour]

Peer Review Information

Journal: Nature Human Behaviour

Manuscript Title: Negativity drives online news consumption

Corresponding author name(s): Jay J. Van Bavel and Stefan Feuerriegel

Editorial Note:

This paper represents a combination of two Stage 1 Registered Reports that were originally submitted to Nature Human Behaviour concurrently but independently. Jay Van Bavel, Claire Robertson, and Philip Pärnamets submitted the paper “If it bleeds, it leads’: Negative words in headlines elicit greater engagement with news stories” (referred to as “manuscript 1” in this peer review file). Stefan Feuerriegel, Kaoru Schwarzenegger, and Nicolas Pröllochs submitted the paper “Emotions drive online news consumption: Evidence from large-scale field experiments” (referred to as “manuscript 2”). These manuscripts were reviewed separately by the same 3 reviewers. Following review, we contacted the authors of both manuscripts and asked whether they would be willing to combine forces to produce a single joint manuscript. On receiving author agreement, we issued separate initial decision letters for each manuscript and asked that the authors address both sets of reviews in their joint revision. All subsequent submissions involved a single manuscript.

Reviewer Comments & Decisions:

Decision Letter, initial version, manuscript 1:
--

26th May 2021

Dear Jay,

Thank you once again for your manuscript, entitled “If it bleeds, it leads’: Negative words in headlines elicit greater engagement with news stories,” and for your patience during the peer review process.

Your manuscript has now been evaluated by 3 reviewers, whose comments are included at the end of this letter. Although the reviewers find your protocol to be of interest, they also raise some important concerns. We are very interested in the possibility of proceeding further with your submission in Nature Human Behaviour, but would like to consider your response to these concerns in the form of a revised manuscript before we make a decision on in principle acceptance and Stage 2 submission.

We invite you to revise your Stage 1 Registered Report taking into account reviewer and editor comments. Please highlight all changes in the manuscript text file.

* Include a "Response to reviewers" document detailing, point-by-point, how you addressed each referee comment. If no action was taken to address a point, you must provide a compelling argument. This response will be sent back to the reviewers along with the revised manuscript.

* Ensure that you use our template for Stage 1 Registered Reports to prepare your revised manuscript: https://www.nature.com/documents/NHB_Template_RR_Stage1.docx. Failure to ensure that your revised Stage 1 submission meets our requirements as specified in the template will result in your submission being returned to you, which will delay its consideration.

* In your cover letter, please include the following information:

--An anticipated timeline for completing the study if your Stage 1 submission is accepted in principle.

--A statement confirming that you agree to share your raw data, any digital study materials, computer code (if relevant), and laboratory log for all eventually published results.

--A statement confirming that, following Stage 1 in principle acceptance, you agree to register your approved protocol on the Open Science Framework (<https://osf.io/>) or other recognised repository, either publicly or under private embargo, until submission of the Stage 2 manuscript.

--A statement confirming that if you later withdraw your paper, you agree to the Journal publishing a short summary of the pre-registered study under a section Withdrawn Registrations.

[REDACTED]

We hope to receive your revised manuscript within four to eight weeks. If you cannot send it within this time, please let us know. We will be happy to consider your revision so long as nothing similar has been accepted for publication at Nature Human Behaviour or published elsewhere.

Nature Human Behaviour is committed to improving transparency in authorship. As part of our efforts in this direction, we are now requesting that all authors identified as 'corresponding author' on published papers create and link their Open Researcher and Contributor Identifier (ORCID) with their account on the Manuscript Tracking System (MTS), prior to acceptance. ORCID helps the scientific community achieve unambiguous attribution of all scholarly contributions. You can create and link your ORCID from the home page of the MTS by clicking on 'Modify my Springer Nature account'. For more information please visit www.springernature.com/orcid.

Sincerely,
Aisha

Aisha Bradshaw
Editor
Nature Human Behaviour

Reviewer expertise:

Reviewer #1: communication, emotion

Reviewer #2: computational social science, Registered Reports

Reviewer #3: online communication

Reviewers' Comments:

Reviewer #1:
Remarks to the Author:

The authors present an interesting study with a unique dataset. Although the negativity bias is very well established also for news headlines and online news, it might be worth investigating this with the unique dataset the authors have access to.

The authors justify why another study about negativity bias is justified. Given the many studies on the topic, also several recent ones such the ones below, the authors might want to justify the originality of this study even more. It seems the study is mainly innovative from a methodological point of view, but might not contribute as much to theory.

<https://www.tandfonline.com/doi/full/10.1080/15205436.2020.1782432>
<https://journals.sagepub.com/doi/full/10.1177/0093650217719596>

The analyses seem well conducted. Considering the size of the dataset, even very small and irrelevant effects might become significant. It is therefore important that the authors interpret the effect sizes of their findings thoroughly.

The authors use multilevel modeling. I am wondering how the data is nested (what are the levels?), and whether it is somehow controlled for unique users (not only unique clicks for one article headline but for users taking part in many of the RTCx).

Sentiment analysis: it would be great if the authors elaborate more on the procedure. Was the sentiment analysis only based on single words? So if a heading contained of a positive and a negative word this heading counted for both? How were negations taken into account ("no fun at all")? Do we know how accurately the sentiment analyses reflect the emotionality of the headlines? Was this compared to manual coding of a subsample?

I am wondering whether images were presented along the headlines, and whether these could have had a valence on their own?

Reviewer #2:
Remarks to the Author:

This paper uses a rich news article RCT dataset to provide causal evidence for negativity bias in news engagement. They focus on news articles with the same body and headlines that vary in the number of emotion words used. Using this data, they are able to show negativity bias (or rather the effect of the presence of negative words in the article headline) even accounting for other factors (e.g. topic).

The dataset is unique and a good fit for the problem studied. The models used are appropriate. However, I do have some concerns.

My first concern relates to the central claim made in this paper. The authors claim in numerous places that the evidence in past work related to negativity bias is limited to in lab studies. This is not true. The authors are missing important scholarship in the field. There are a number of studies that use social media data and online news to provide evidence for negativity bias (e.g. Berger et al. "What makes online content viral?" for news on social media, Stieglitz, S. & Dang-Xuan, L. "Emotions and information diffusion in social media: Sentiment of microblogs and sharing behavior." for an example of this in political communication and Naveed, Nasir, Thomas Gottron, Jérôme Kunegis, and Arifah Che Alhadi. "Bad news travel fast: A content-based analysis of interestingness on twitter for general social media). There are also studies that suggest stronger emotion—either positive or negative—can be associated with stronger engagement (e.g., Jenders M, Kasneci G, Naumann F: Analyzing and predicting viral tweets.) While these studies are not providing causal claims, the authors' claim that only evidence for negativity bias is coming from lab experiments is not true. I would recommend authors to tone down the claims and provide a more comprehensive summary of the scholarship in the field.

I also would appreciate more insights about the kinds of words being identified by the different dictionaries and whether they indeed correspond to human notions of emotions. There are increasingly more studies that question these dictionary-based approaches (e.g. Kross, Ethan, Philippe Verduyn, Margaret Boyer, Brittany Drake, Izzy Gainsburg, Brian Vickers, Oscar Ybarra, and John Jonides. "Does counting emotion words on online social networks provide a window into people's subjective experience of emotion? A case study on Facebook.") so I think the authors would benefit from some discussion here. It would also be helpful to see the list of most common negative and positive words and to see them in the context of some real headlines in the dataset. Examples of headlines can help the reader better understand the nature of the treatment. While the paper provides convincing evidence for the relationship between the use of nrc dictionary words and engagement, I think the paper would benefit from further evidence that such words indeed correspond to/trigger emotions.

Finally, the writing could be improved. For instance, the exact structure of the Upworthy dataset and the use of registered reports was not clear to me after reading the manuscript. I got a clearer sense after visiting the Upworthy dataset website. Added clarity would help future reviewers.

Reviewer #3:

Remarks to the Author:

The proposed research, “If it bleeds, it leads”: Negative words in headlines elicit greater engagement with news stories’, seeks to investigate the effect of negativity on the click-through rate of news headlines. To that end, the authors propose to use the full Upworthy Research Archive and present their planned methodology using a small subset of the data. The research question is certainly relevant for a broad, multidisciplinary audience and I concur that the dataset might provide compelling answers, some limitations (see below) notwithstanding.

This review presents somewhat of a novelty to me, as I was asked by the journal to review two similar stage 1 registered reports by different authors. Both proposals focus on the effects of emotion, especially negative emotion, on click-through rates. Both proposals use the Upworthy Research Archive. Both proposals share the main preliminary conclusion in that negativity indeed seems to causally increase news click-through rate.

Both proposals also share their biggest weakness in that they rely on (different) off-the-shelf sentiment dictionaries, but fail to (re-)validate them for the specific context they are used in. Validation, of course, is absolutely crucial in the context of automated text analysis: “The validity of a method or tool is dependent on the context in which it is used, so even if a researcher uses an existing off-the-shelf tool with published validity results it is vital to show how well it performs in a specific domain and on a specific task.”^{1–3} A manual validation of the used dictionary on the dataset the analyses are performed on—Song et al. recommend 1% of the data⁴—should thus demonstrate that the bag-of-words and additivity assumptions (more negative than positive words = negative sentiment, and the bigger the difference, the more negative) holds true and that the dictionary can be used on the thematic domain of the data.

Regarding the first point, the additivity assumption is explicitly modelled in the paper at hand as the amount of negative words serves as a predictor of clicking on an article. The authors implicitly address an often-discussed critique of sentiment dictionaries of just adding up negative and positive terms to form a sentiment score by modelling positive and negative words independently. This, however, creates a subtle but crucial change in meaning, as by allowing headlines to be both negative and positive at the same time, the authors do not, as written in the introduction, examine whether “negative headlines increase engagement” (p. 3), but the effect of negative words (which may very well appear in a ‘positive’ headline). This distinction is never made explicit in the manuscript; in fact, the wording in the introduction (“negative stimuli”, “negative information”, “negative headlines”, “negative news”) seems to suggest that the authors equate the amount of negative words with the overall negative valence of a piece of news.

Consider an extreme example: The certainly positive, made-up headline “Disaster averted: We have conquered war, famine and death” contains only negative (and neutral) words according to the Bing dictionary. This leads to the second point raised above, the domain-specific validity of the dictionary, which is especially relevant given the object of investigation: Upworthy is, by their journalistic self-conception, an outlet for positive news and storytelling and, as such, an unusual choice to study the effects of “negative news” (p. 4). Thus, the authors need to demonstrate with a manual validation that they are indeed able to measure negative sentiment with the Bing dictionary (or any other dictionary, for that matter) in this specific context. (Upworthy may be an ‘outlier’ in another respect as well, in that it had a reputation of being especially ‘click-baity’; as such, users may select news on Upworthy based on different criteria than on, say, more matter-of-fact oriented news outlets).

The unique situation of reviewing two very similar research proposals at the same time provides me with the opportunity to not only review each proposal on its own, but also to look at relative (dis-)advantages. While sharing some of the validation problems outlined above, the 'other' submission does explicitly and both theoretically and methodologically address the tenuous connection between negative words and negative sentiment; it also provides several additional robustness checks (e.g., negations of words) and contextualizes the results with the topics and themes of the articles in the dataset.

1. van Atteveldt, W. & Peng, T.-Q. When communication meets computation: Opportunities, challenges, and pitfalls in computational communication science. *Commun. Methods Meas.* 12, 81–92 (2018).
2. Grimmer, J. & Stewart, B. M. Text as Data: The Promise and Pitfalls of Automatic Content Analysis Methods for Political Texts. *Polit. Anal.* 21, 267–297 (2013).
3. Chan, C. et al. Four best practices for measuring news sentiment using 'off-the-shelf' dictionaries: a large-scale p-hacking experiment. <https://osf.io/np5wa> (2020) doi:10.31235/osf.io/np5wa.
4. Song, H. et al. In validations we trust? The impact of imperfect human annotations as a gold standard on the quality of validation of automated content analysis. *Polit. Commun.* 1–23 (2020) doi:10.1080/10584609.2020.1723752.

Decision Letter, initial version, manuscript 2:

26th May 2021

Dear Stefan,

Thank you once again for your manuscript, entitled "Emotions drive online news consumption: Evidence from large-scale field experiments," and for your patience during the peer review process.

Your manuscript has now been evaluated by 3 reviewers, whose comments are included at the end of this letter. Although the reviewers find your protocol to be of interest, they also raise some important concerns. We are very interested in the possibility of proceeding further with your submission in *Nature Human Behaviour*, but would like to consider your response to these concerns in the form of a revised manuscript before we make a decision on in principle acceptance and Stage 2 submission.

We invite you to revise your Stage 1 Registered Report taking into account reviewer and editor comments. Please highlight all changes in the manuscript text file.

* Include a "Response to reviewers" document detailing, point-by-point, how you addressed each referee comment. If no action was taken to address a point, you must provide a compelling argument. This response will be sent back to the reviewers along with the revised manuscript.

* Ensure that you use our template for Stage 1 Registered Reports to prepare your revised manuscript: https://www.nature.com/documents/NHB_Template_RR_Stage1.docx. Failure to ensure that your revised Stage 1 submission meets our requirements as specified in the template will result in your submission being returned to you, which will delay its consideration.

* In your cover letter, please include the following information:

--An anticipated timeline for completing the study if your Stage 1 submission is accepted in principle.

--A statement confirming that you agree to share your raw data, any digital study materials, computer code (if relevant), and laboratory log for all eventually published results.

--A statement confirming that, following Stage 1 in principle acceptance, you agree to register your approved protocol on the Open Science Framework (<https://osf.io/>) or other recognised repository, either publicly or under private embargo, until submission of the Stage 2 manuscript.

--A statement confirming that if you later withdraw your paper, you agree to the Journal publishing a short summary of the pre-registered study under a section Withdrawn Registrations.

[REDACTED]

We hope to receive your revised manuscript within four to eight weeks. If you cannot send it within this time, please let us know. We will be happy to consider your revision so long as nothing similar has been accepted for publication at Nature Human Behaviour or published elsewhere.

Nature Human Behaviour is committed to improving transparency in authorship. As part of our efforts in this direction, we are now requesting that all authors identified as 'corresponding author' on published papers create and link their Open Researcher and Contributor Identifier (ORCID) with their account on the Manuscript Tracking System (MTS), prior to acceptance. ORCID helps the scientific community achieve unambiguous attribution of all scholarly contributions. You can create and link your ORCID from the home page of the MTS by clicking on 'Modify my Springer Nature account'. For more information please visit please visit www.springernature.com/orcid.

Sincerely,
Aisha

Aisha Bradshaw
Editor
Nature Human Behaviour

Reviewer expertise:

Reviewer #1: communication, emotion

Reviewer #2: computational social science, Registered Reports

Reviewer #3: online communication

Reviewers' Comments:

Reviewer #1:

Remarks to the Author:

The authors present an interesting study with a unique dataset. I believe that testing the effects of emotional sentiment in news headlines is important and will be of interest to many readers.

My comments mainly relate to the theoretical framework that was used in this study.

1. I am not so convinced by the emotion model (Plutchick) that was used as a background for this study. First, I am not sure in how far this is still an established model, as it seems to be less frequently used in the scientific literature than other emotion models. Most importantly, I am not sure if I agree with one of the main premises, namely that anger is a positive emotion. I think, traditionally, anger has been conceptualized as a negative emotion. Anger might have positive components or can be beneficial for the person showing this emotion, but I think that the general notion is that anger is a negative emotion. I would therefore rather see the findings when anger is coded as a negative emotion. Or I need a more compelling argument for using anger as a positive emotion.

Moreover, I am not convinced by Plutchik's dyads. Can the authors elaborate more on why using this model and not another emotion model?

Related to this, concerning the methodology, what is the advantage of the bipolar emotions, and why were they calculated as mentioned on page 10-11?

Dyads: the reasoning for using the dyads is also not clear to me, what does this add? Why not just investigate the differences between the 8 basic emotions, or just positive vs negative?

2. The authors state in the theory section that "randomized experiments have found that estimates of the influence of online media....can be incorrect by more than 100%". Can the author further clarify this sentence? What types of estimates, and in which/how many studies has this been found (and which topic)?

3. The authors argue that "if negative emotions drive news consumption, the same negative emotions are likely to elicit negative emotions among readers." I think this is somewhat overstated, and not necessarily true. If I read a negative message about a political opponent, I might feel more positive instead of negative. Also the sensational value of negative news can have positive effects on audiences (e.g., entertainment, enjoyment) rather than negative effects (for example true crime series are popular and perceived as entertaining and enjoyable).

-
4. The authors mention that "a better understanding of the emotion effect on news consumption can inform how to curb the proliferation of misinformation." Maybe the authors can elaborate on this a bit more. How can an understanding of the fact that negative news are clicked on more frequently curb the proliferation of misinformation?
-
5. Is the gunning-fox index still used and accepted?
6. Question about the CTR: are these somehow controlled for the same persons being part of several RTC? Are the data nested? Do we know how many users did participate in these trials?
7. Does the sentiment analysis take negations into account, as in the example ("not amused" indicating irritation rather than amusement)? Do we know how accurately the sentiment analyses reflect the emotionality of the headlines? Was this compared to manual coding of a subsample?

Reviewer #2:

Remarks to the Author:

This paper uses a rich dataset of news article RCT dataset to study the relationship between emotions and news engagement. Use of the Upworthy RCT dataset allows the authors to identify the effect of emotions embedded in the news articles while accounting for other confounding factors such as the body of the article. The authors use Plutchik's emotion model, which identifies a comprehensive set of basic emotions as well as more complex ones that are derived from them.

While I found aspects of the paper interesting, I have some concerns.

First, I appreciated that the authors provided example headlines for the reader to inspect (table 1). However, inspecting it, I am more concerned that the dictionary based approach might not be corresponding to human perception of emotions. For instance, the 4 sentences are positive, neutral, negative, and neutral, if I didn't miscount the colored words. But they look rather comparable. There are some recent studies that question the use of emotion dictionaries (eg. Kross, Ethan, Philippe Verduyn, Margaret Boyer, Brittany Drake, Izzy Gainsburg, Brian Vickers, Oscar Ybarra, and John Jonides. "Does counting emotion words on online social networks provide a window into people's subjective experience of emotion? A case study on Facebook"). It would be great if the authors can perform some sort of validation test here, perhaps getting human judgment on the emotions conveyed by the headlines, perhaps gathered through crowdsourcing platforms etc.

Second concern I have is the number of models/effects tested. While it is interesting that the emotion model allowed the researchers to delve deeper to provide a deeper look, it also led them to testing various hypotheses. It would be best if the authors can perform multiple hypothesis testing correction.

Third, it would be great to get more insights about the RCTs that created the data used in this paper. Authors mention that there are multiple packages with identical headlines for instance. Not sure why. I would also be curious to learn more about whether the RCTs were generally varying some other factor (say, sarcasm). These details maybe don't need to be in the main manuscript but more insights about the data would help.

The sentiment model: the authors had mentioned in their discussion of related work that some studies find stronger engagement for content with stronger sentiment (e.g. very positive and very negative). I am curious why they went with a linear model for sentiment at the end (formula 3). There are already various models tested so perhaps it is not ideal to add but curious about the decision process that led to excluding this as an alternative explanation to test.

Topic modeling: The topic model is not evaluated. As such, I find it hard to take the results at face value. Topic models commonly fail in document assignment despite top-k words looking good (table-6 has decent top-10 words). I would encourage the authors to validate this model before using. Here, again, they can rely on human judgment on a subset of headlines. The authors might also consider using the body of the article in performing this topic analysis.

Minor note: I am not an emotions researcher. I found it surprising that anger was classified under positive sentiment. Is this widely accepted or only in the Plutchik model?

Reviewer #3:

Remarks to the Author:

The proposed research, 'Emotions drive online news consumption: Evidence from large-scale field experiments,' seeks to investigate the effect of emotions on the click-through rate of news headlines. To that end, the authors propose the use of the full Upworthy Research Archive and present their planned methodology using a small subset of the data. The research question is certainly relevant for a broad, multidisciplinary audience and I concur that the dataset might provide compelling answers, some limitations (see below) notwithstanding.

This review presents somewhat of a novelty to me, as I was asked by the journal to review two similar stage 1 registered reports by different authors. Both proposals focus on the effects of emotion, especially negative emotion, on click-through rates. Both proposals use the Upworthy Research Archive. Both proposals share the main preliminary conclusion that negativity indeed seems to causally increase news click-through rates.

Both proposals also share their biggest weakness in that they rely on (different) off-the-shelf sentiment dictionaries, but fail to (re-)validate them for the specific context they are used in. Validation, of course, is absolutely crucial in the context of automated text analysis: "The validity of a method or tool is dependent on the context in which it is used, so even if a researcher uses an existing off-the-shelf tool with published validity results it is vital to show how well it performs in a specific domain and on a specific task." 1-3 A manual validation of the used dictionary on the dataset the analyses are performed on—Song et al. recommend 1% of the data⁴—should thus demonstrate that the bag-of-words and additivity assumptions (more negative than positive words = negative sentiment, and the bigger the difference, the more negative) holds true and that the dictionary can be used on the thematic domain of the data.

Regarding the first point, the additivity assumption is explicitly used in the calculation of overall negative vs. positive sentiment as well as in the calculations of bipolar emotions. The validity of this assumption is especially crucial in the context of the NRC dictionary where the same words can load onto both

'negative' and 'positive' emotions (which leads to some peculiarities, such as the word 'suicide' being coded as overall neutral, whereas 'giant' is an overall negative word, see p. 36; Table 3 also suggests that 'lying' is considered to be an emotional word in the present tense, but not in the past tense). Second, the domain-specific validity of the dictionary is especially relevant given the object of investigation: Upworthy is, by their journalistic self-conception, an outlet for positive news and storytelling and, as such, an unusual choice to study the effects of "negative emotions" (p. 6). Thus, the authors need to demonstrate that the NRC is indeed able to adequately measure the eight emotions and negative/positive sentiment in the context of Upworthy. For example, to use the examples provided by the authors, they need to show that "Some College Kids Figured Out How To Cut Energy Companies Out Of Our Lives. They Are Not Amused." is indeed perceived to be more positive than "Some Young Punks Just Punked A Giant Corporation Out Of The Easy Money It Was Bilking Off You", which in turn is perceived to be more positive than "Note To Giant Corporation: Don't Mess With Young Punks Who Have More Brains Than You Have Money", with each headline leading to the appraisal of the associated emotions. This validation is especially relevant in the context of comparatively short headlines where a single word might substantially alter the coded sentiment.

One additional limitation of the dataset is that Upworthy had a reputation of being especially 'click-baity'; users were thus likely to expect emotional wording in the headlines, which might affect the representativeness of Upworthy user behavior for all online news.

The unique situation of reviewing two very similar research proposals at the same time provides me with the opportunity to not only review each proposal on its own, but also to look at relative (dis-)advantages. While sharing some of the validation problems outlined above, the project at hand provides a more granular approach, contains additional robustness checks (e.g., negations of words) and contextualizes the results with the topics and themes of the articles in the dataset (however, topic models should be manually validated as well, for example with a word intrusion test). Thus, should the editors decide to move forward with only one of the submitted projects, I would suggest it to be this one. However, the validity of the automatic emotion coding, as outlined above, needs to be established before an acceptance in principle.

1. van Atteveldt, W. & Peng, T.-Q. When communication meets computation: Opportunities, challenges, and pitfalls in computational communication science. *Commun. Methods Meas.* 12, 81–92 (2018).
2. Grimmer, J. & Stewart, B. M. Text as Data: The Promise and Pitfalls of Automatic Content Analysis Methods for Political Texts. *Polit. Anal.* 21, 267–297 (2013).
3. Chan, C. et al. Four best practices for measuring news sentiment using 'off-the-shelf' dictionaries: a large-scale p-hacking experiment. <https://osf.io/np5wa> (2020) doi:10.31235/osf.io/np5wa.
4. Song, H. et al. In validations we trust? The impact of imperfect human annotations as a gold standard on the quality of validation of automated content analysis. *Polit. Commun.* 1–23 (2020) doi:10.1080/10584609.2020.1723752

Author Rebuttal to Initial comments

Response letter for collaborative manuscript:

“Negativity drives online news consumption: Evidence from large-scale field experiments”

Dear Dr. Bradshaw & reviewers,

Thank you for the opportunity to revise and resubmit our papers, "*If it bleeds, it leads: Negativity bias in the news*" and "*Emotions drive online news consumption: Evidence from large-scale field experiments*." We combined both in a collaborative manuscript "*Negativity drives online news consumption: Evidence from large-scale field experiments*." The two analyses had found evidence of negativity bias using different approaches across four lexicons. Importantly, across both papers, we arrive at consistent findings.

The unique opportunity to collaborate on a revised manuscript, incorporating suggestions from the reviews of both papers, has made, in our opinion, the combined manuscript much stronger. The submitted manuscript represents the best efforts from a dedicated team of researchers. For thoroughness, both author teams have responded directly to the reviewers of their original manuscript.

Before delving into specific comments from reviewers for each manuscript, we wanted to address some of the overarching feedback and how we improved our paper:

- **Combined storyline on negativity bias.** In combining the two manuscripts, we merged our two separate storylines (negativity bias vs. emotions) as follows. (1) We decided to keep the narrative around a negative bias as our main research theme. Specifically, we examine negativity bias in online news consumption based on a unique dataset (Upworthy Research Archive). We use preregistered hypotheses that are carefully derived from literature and for which we report large-scale causal evidence from field experiments. (2) We further conduct a secondary "exploratory" analysis in which we analyze the role of different emotions. This analysis should be seen as an "exploratory" analysis to provide richer insights.
- **New user studies validating dictionary approach** All six reviews encouraged us to revalidate the dictionaries we used for our specific context. We thought this was a very helpful suggestion, and have therefore conducted new user validation studies in accordance with best practices (e.g., Song et al. 2018). The user studies are reported in a new Supplement D. Study 1 validates that users perceive headlines with negative words as more negative. Study 2 validates that users perceive headlines with discrete emotional words (such as angry words) as reflecting that emotion. This confirms that our dictionary approach captures the actual perceptions of emotions.

We asked participants whether they thought the headline they were reading was more negative

or positive, not whether the headline made them feel more negative or positive.

This is an important distinction, as most studies on negativity bias to this point have focused on the subjective experience of participants (i.e., Kross et al. 2019). Our study is focused on how emotional rhetoric changes people's quantifiable behaviour (i.e., news consumption).

- **New user studies validating topic model.** We took inspiration in (Brady et al., 2017) and performed an additional user study to validate our topic model (see new materials in Supplement H.3). For this, we followed the approach in (Chang et al., 2009), which is considered best-practice for validating topic models. The objective behind our user study is to validate that participants were better than chance at categorizing headlines as belonging to a certain topic in accordance with our topic models ("topic intrusion"). The new and extensive analysis strengthens the validity of our findings. Taken together, both user studies confirm that the topic model combines news headlines into meaningful topics.

We believe the new analyses address the main suggestions in both reviewer teams. We thank the reviewers for their kind words and constructive feedback on both papers. Responses to specific comments (*in italics*) can be found below.

Response letter for manuscript #1: "If it bleeds, it leads: Negativity bias in the news"

Manuscript #1 / Responses to Reviewer 1

Comment R1.1: "The authors present an interesting study with a unique dataset. Although the negativity bias is very well established also for news headlines and online news, it might be worth investigating this with the unique dataset the authors have access to."

Response: Thank you for the positive feedback!

Comment R1.2: "The authors justify why another study about negativity bias is justified. Given the many studies on the topic, also several recent ones such the ones below, the authors might want to justify the originality of this study even more. It seems the study is mainly innovative from a methodological point of view, but might not contribute as much to theory."

<https://www.tandfonline.com/doi/full/10.1080/15205436.2020.1782432>

<https://journals.sagepub.com/doi/full/10.1177/0093650217719596>

Response: We agree that there have been several studies, including recent work, examining negativity bias, including negativity bias in the news. The proposed registered report is original in three ways:

- (1) *Rigor*. The unique nature of the data set allows for an examination of real-world news consumers that is also causal. All findings are based on $N = 22,743$ randomized controlled trials for a platform with more than 500 million impressions, thus providing large-scale evidence. In our views, such data are practically unheard of in social psychology. Hence, the sample size is larger than the vast majority of studies on this topic. We apologize if we failed to make this more clear in our initial submission and hope the current version of the paper is more clear about these substantive contributions.
- (2) *Research question*. We study emotions as a driver of online news consumption. This is different from other research that studies emotions in sharing (e.g., Stieglitz et al 2013, , Berger & Milkman 2012; Vosougi et al 2018; Chaui & Zhao 2020), as opposed to news consumption.
- (3) *Theory*. We not only examine the presence of a negativity bias but, in an additional analyses, further study other, complex emotions.

We have added the two suggested references from above to the manuscript. When combining our papers into a collaborative one, we have revised our text and further strengthened the differences between prior literature and our work, highlighting how our paper is original.

Comment R1.3: "The analyses seem well conducted. Considering the size of the dataset, even very small and irrelevant effects might become significant. It is therefore important that the authors interpret the effect sizes of their findings thoroughly."

Response: We expanded our passages on how to interpret the effect sizes in our dataset. By combining the two manuscripts, we have incorporated several improvements. (1) We opted for a visual display of the estimation results that help readers focus on the effect sizes (rather than p -values or significance levels). (2) We additionally reported the odds as an intuitive way how changes in emotion affect click-through rates. (3) We revised our results section to interpret effect sizes more thoroughly. For example, we included statements such as "For each negative word in a headline, the likelihood that a person would click on that headline increased by 1.0%."

Comment R1.4: "The authors use multilevel modeling. I am wondering how the data is nested (what are the levels?), and whether it is somehow controlled for unique users (not only unique

clicks for one article headline but for users taking part in many of the RTCx).

Response: We agree that this is an important feature of the data to be clear to readers. Our data will consist of $N = 22,743$ different news stories (levels) each associated with its own randomized controlled trial (RCT). For each news story (i.e., for each RCT), around 4.29 headline variations from the same news story were compared. Following your comment, we have added more information about the structure of the data, as well as a graphic (see new Figure 1 in the main manuscript, new Supplement B, and new Supplement C). We are happy to add more details if there is anything unclear.

Comment R1.5: "Sentiment analysis: it would be great if the authors elaborate more on the procedure. Was the sentiment analysis only based on single words? So if a heading contained of a positive and a negative word this heading counted for both? How were negations taken into account ("no fun at all")? Do we know how accurately the sentiment analyses reflect the emotionality of the headlines? Was this compared to manual coding of a subsample?"

Response: Thank you for raising these relevant questions, based on which we have revised our paper in the following ways:

- (1) **Details on sentiment analysis.** We extended and clarified the procedure of our sentiment analysis (see revised section "Methods"). Specifically, our procedure follows existing social science research (Brady et al. 2017; Vosoghi et al 2018; Rathje et al. 2021). Sentiment analysis was based on single words ("unigrams") due to the short length of the headlines (mean length: 14.89 words). We then counted the number of positive words (n_{positive}) and the number of negative words (n_{negative}) in each headline. Here, a word is considered 'positive' if it is in the dictionary of positive words (and, vice versa, for negative words). We then normalize the frequency by the length of the headline, that is, the total number of words in the headline (n_{total}). This yields the two sentiment scores

$$Positive_{ij} = \frac{n_{\text{positive}}}{n_{\text{total}}} \text{ and } Negative_{ij} = \frac{n_{\text{negative}}}{n_{\text{total}}},$$

which are used in the regression. If a heading has 10 words out of which one is classified as 'positive' and none as 'negative', the scores are $Positive_{ij} = 10\%$ and $Negative_{ij} = 0\%$. If a heading has 10 words and contains one 'positive' and one 'negative' word, the scores are $Positive_{ij} = 10\%$ and $Negative_{ij} = 10\%$.

- (2) **Negation handling.** We included negation handling into our text mining framework. This is now—by default—part of the main analysis. As such, we reverse the sentiment direction

when a negation is present. For example, the phrase “not fun” would be coded as negative rather than positive due to the negation. More formally, the following procedure is used. First, the text is scanned negation terms using a predefined list, and then all emotional words in the neighborhood are counted as belonging to the opposite sentiment (or emotion). In our analysis, the neighborhood is set to 5 words before and 2 words after the negation. Details are in the revised “Methods”.

We also added a new analysis comparing the above approach to negation handling based on an alternative neighborhood (i.e., the so-called negation scope) to confirm the robustness of our findings. Both analyses led to qualitatively identical results (see new Supplement F.2).

- (2) **Validation based on user study.** To ensure that our dictionary analysis captured emotionality in headlines, we performed two new user studies to validate the accuracy of our dictionary approach (see new Supplement D). Study 1 validates that user’s judgments of emotionality in headlines correlate with the number of negative and/or positive words each headline contains. We found user judgments of sentiment and our sentiment scores were significantly positively correlated at $r_s = .36$ ($p < .001$). These findings validate that our dictionary approach captures significant variation in the perception of emotions in headlines from users. Study 2 validates that users ($k = 10$ as in Song et al. 2020) perceive headlines with discrete emotional words (such as angry words) as reflecting that emotion. We found that the overall correlation between NRC dictionary scores and user judgments for the 8 discrete emotions was positive and statistically significant ($r_s = 0.13$; $p < 0.001$). Detailed results for each dimension of the NRC emotion lexicon are in the new Supplement D. This validates that our dictionary approach captures the perceptions of emotions from users.

Comment R1.6: “I am wondering whether images were presented along the headlines, and whether these could have had a valence on their own?”

There were varying images with the headlines that could have had valence of their own. Unfortunately, the Upworthy research archive does not make image data available to researchers. We have added more details about the features of each headline that can vary. Critically, the negativity bias remains robust even when only headlines with no image test are included in the analysis suggesting that images can not account for the overall pattern of findings (see new Supplement F.6). The same analysis will be performed on the full data set, where the sample size will be larger to ensure the images are not driving the effects.

Manuscript #1 / Responses to Reviewer 2

Comment R2.1: “This paper uses a rich news article RCT dataset to provide causal evidence for negativity bias in news engagement. They focus on news articles with the same body and headlines that vary in the number of emotion words used. Using this data, they are able to show negativity bias (or rather the effect of the presence of negative words in the article headline) even accounting for other factors (e.g. topic).”

Response: Thank you for the positive feedback on our manuscript!

Comment R2.2: “The dataset is unique and a good fit for the problem studied. The models used are appropriate. However, I do have some concerns.”

Response: Thank you. We detail below point-by-point how we improved our paper based on your suggestions.

Comment R2.3: “My first concern relates to the central claim made in this paper. The authors claim in numerous places that the evidence in past work related to negativity bias is limited to in lab studies. This is not true. The authors are missing important scholarship in the field. There are a number of studies that use social media data and online news to provide evidence for negativity bias (e.g. Berger et al. “What makes online content viral?” for news on social media, Stieglitz, S. & Dang-Xuan, L. “Emotions and information diffusion in social media: Sentiment of microblogs and sharing behavior.” for an example of this in political communication and Naveed, Nasir, Thomas Gottron, Jérôme Kunegis, and Arifah Che Alhadi. “Bad news travel fast: A content-based analysis of interestingness on twitter for general social media). There are also studies that suggest stronger emotion—either positive or negative—can be associated with stronger engagement (e.g., Jenders M, Kasneci G, Naumann F: Analyzing and predicting viral tweets.) While these studies are not providing causal claims, the authors’ claim that only evidence for negativity bias is coming from lab experiments is not true. I would recommend authors to tone down the claims and provide a more comprehensive summary of the scholarship in the field.”

Response: Thank you for the references. We agree that there have been several studies (e.g., Berger et al 2012, Stieglitz et al 2013, etc.) examining negativity bias based on observational data. We have narrowed our claims regarding past work, and improved our review of literature in the field by including the citations recommended by the reviewer, as well as additional ones (e.g., Kim et al. 2012; Chuai & Zhao 2020; Zollo 2015). We are happy to include additional relevant work if you see any further omissions.

Furthermore, we have increased the justification of the originality of the proposed Registered Report. The proposed Registered Report is original due to the unique nature of the dataset, which allows for an examination of real-world news consumers that is also causal, data practically unheard of in social psychology. Moreover, the sample size is much larger than the vast majority of work on this topic.

Comment R2.4: "I also would appreciate more insights about the kinds of words being identified by the different dictionaries and whether they indeed correspond to human notions of emotions. There are increasingly more studies that question these dictionary-based approaches (e.g. Kross, Ethan, Philippe Verduyn, Margaret Boyer, Brittany Drake, Izzy Gainsburg, Brian Vickers, Oscar Ybarra, and John Jonides. "Does counting emotion words on online social networks provide a window into people's subjective experience of emotion? A case study on Facebook.") so I think the authors would benefit from some discussion here. It would also be helpful to see the list of most common negative and positive words and to see them in the context of some real headlines in the dataset. Examples of headlines can help the reader better understand the nature of the treatment."

Response: We agree that readers would benefit from more transparency regarding the type of words the text mining analysis is examining. We have thus improved our manuscript in the following ways:

- (1) **List of common sentiment/emotion words.** We have added a table with the most common negative and positive words (see new Supplement C). We also list a similar table with the most common emotional words.
- (2) **Differences from Kross et al.** Furthermore, we agree that dictionary-based approaches have significant limitations for inferring the subjective state of emotion. However, our work differs in several important ways from the critique presented in Kross et al. 2020. First, we are not claiming that emotional words in headlines are corresponding with a news reader's actual emotion. For example, it is conceivable that an angry headline about an outgroup member would create joyful feelings in a reader. However, our research is silent about what participants' experienced emotions are, and is solely concerned with examining the type of language that increases news consumption. Secondly, the paper cited above refers to interpreting an individual's emotions based on participant's own social media posts. Importantly, Upworthy.com was not a social media website, and we are not analyzing readers' comments or even their sharing -- we are only analyzing their decision to click on a given headline. This is an important clarification which we have improved in the manuscript in its current Stage 1 state, and are happy to include these clarifications in a discussion section at the Registered Report stage 2. We hope that this clarifies our specific research question

and differentiates it from much of the prior work on related issues.

Comment R2.5: "While the paper provides convincing evidence for the relationship between the use of nrc dictionary words and engagement, I think the paper would benefit from further evidence that such words indeed correspond to/trigger emotions."

Response: We agree that our original analyses could not adequately answer the question of whether headlines with more negative words were perceived as more negative. To address this concern, we performed two user studies to validate the accuracy of our dictionary approach (see new Supplement D). Study 1 validates that users ($k = 10$) perceive headlines with negative words as being more negative. Study 2 validates that users ($k = 10$) perceive headlines with discrete emotional words (such as angry words) as reflecting that emotion. This confirms that our dictionary approach captures the actual perceptions of emotions.

As an important note, we asked participants whether they thought the headline they were reading was more negative or positive, not whether the headline made them feel more negative or positive. This is an important distinction, as most studies on negativity bias to this point have focused on the subjective experience of participants (i.e., Kross et al. 2019). Our study, on the other hand, is ambivalent towards the reader's emotional experience. We are only making claims based on participants' quantifiable behaviour (i.e., news consumption).

Comment R2.6: "Finally, the writing could be improved. For instance, the exact structure of the Upworthy dataset and the use of registered reports was not clear to me after reading the manuscript. I got a clearer sense after visiting the Upworthy dataset website. Added clarity would help future reviewers."

Response: We agree that both the format of a Registered Report and the Upworthy dataset deserves more explanation. To provide more details, we expanded our manuscript in the following ways:

- (1) We added an overview describing the general procedure of a Registered Report. The new overview is provided on p. 3 (Procedure of "Registered Report").
- (2) We added further explanations on the role of Upworthy as an online news platform and the specifics of our Upworthy dataset. In particular, we elaborate more clearly on what makes the dataset unique and how the different randomized controlled trials were conducted. We included additional figures showing example RCTs and summary statistics.
- (3) We will further improve the structure of the paper in Stage 2 of the Registered Report.

For Stage 1 of the Registered Report, we followed common practice and, e.g., listed the Methods before providing the display items with summary statistics. We understand that this might not aid comprehensibility but it is the predetermined structure for Stage 1. In Stage 2, we are allowed to follow the typical format of papers at *Nature Human Behaviour* where the Methods are at the end. We are confident that this change will help readers navigate through our manuscript.

Manuscript #1 / Responses to Reviewer 3

Comment R3.1: "The proposed research, "If it bleeds, it leads": Negative words in headlines elicit greater engagement with news stories', seeks to investigate the effect of negativity on the click-through rate of news headlines. To that end, the authors propose to use the full Upworthy Research Archive and present their planned methodology using a small subset of the data. The research question is certainly relevant for a broad, multidisciplinary audience and I concur that the dataset might provide compelling answers, some limitations (see below) notwithstanding."

Response: Thank you for seeing value, especially finding it relevant for a broad, multidisciplinary audience.

Comment R3.2: "This review presents somewhat of a novelty to me, as I was asked by the journal to review two similar stage 1 registered reports by different authors. Both proposals focus on the effects of emotion, especially negative emotion, on click-through rates. Both proposals use the Upworthy Research Archive. Both proposals share the main preliminary conclusion in that negativity indeed seems to causally increase news click-through rate."

Response: We agree that it was an interesting opportunity to develop and test research. The unique setting to collaborate jointly on a revised manuscript has led -- in our opinion -- to a much stronger manuscript. Importantly, we found it reassuring to see that two separate analyses with different methodologies and across four different lexicons have arrived at consistent findings. The collaborative manuscript combines the best ideas from both research teams.

Comment R3.3: "Both proposals also share their biggest weakness in that they rely on (different) off-the-shelf sentiment dictionaries, but fail to (re-)validate them for the specific context they are used in. Validation, of course, is absolutely crucial in the context of automated text analysis: "The validity of a method or tool is dependent on the context in which it is used, so even if a researcher uses an existing off-the-shelf tool with published validity results it is vital to show how well it

performs in a specific domain and on a specific task.” 1–3 A manual validation of the used dictionary on the dataset the analyses are performed on—Song et al. recommend 1% of the data⁴—should thus demonstrate that the bag-of-words and additivity assumptions (more negative than positive words = negative sentiment, and the bigger the difference, the more negative) holds true and that the dictionary can be used on the thematic domain of the data.”

Response: Thank you for this helpful suggestion! We agree that our original analyses could not adequately answer the question of whether headlines with more negative words were perceived as more negative. We thus followed best practices (e.g., Song et al. 2018) and performed two additional user studies as validation. The objective behind this was to ensure that our dictionary analysis captured emotionality in headlines. We included $k = 10$ raters, and examined 2.3% of our data in the validation, i.e., 2x the amount recommended by Song et al (2018). We added the two user studies in a new Supplement D. Study 1 validates that users perceive headlines with negative words as more negative. Study 2 validates that users perceive headlines with discrete emotional words (such as angry words) as reflecting that emotion. Across both, we compared the accuracy of the dictionary approach against the judgments of human raters. Overall, we find that our dictionary approach is successful in capturing the actual perceptions of emotions in the thematic domain of the data.

Furthermore, the correlation between user ratings of sentiment and computer calculated sentiment speaks to the additivity assumption. (1) A headline with two negative words is perceived as more negative than a headline with only one negative word, and so on. (2) A headline with two negative words is perceived as more negative than a headline with only one negative word, and so on. (3) A text with more negative than positive words is perceived as having a negative sentiment. (4) A text with more positive than negative words is perceived as having a positive sentiment. Altogether, the computer calculated sentiment has a strong and statistically significant association with sentiment ratings by users, thus implying that our dictionary approach captures the sentiment well.

As an important note, we asked participants whether they thought the headline they were reading was more negative or positive, not whether the headline made them feel more negative or positive. This is an important distinction, as most studies on negativity bias to this point have focused on the subjective experience of participants (i.e., Kross et al. 2019). Our study, on the other hand, is ambivalent towards the reader’s emotional experience. We are only making claims based on participants’ quantifiable behaviour (i.e., news consumption).

When combining the two manuscripts, we found that each relied upon a different sentiment lexicon. We thus incorporated an additional robustness check (see revised Supplement F.1) where we compare the results against alternative sentiment lexicons (LIWC, NRC, and SentiStrength).

This should further demonstrate that, regardless of the lexicon, the conclusions remain robust.

Comment R3.4: “Regarding the first point, the additivity assumption is explicitly modelled in the paper at hand as the amount of negative words serves as a predictor of clicking on an article. The authors implicitly address an often-discussed critique of sentiment dictionaries of just adding up negative and positive terms to form a sentiment score by modelling positive and negative words independently. This, however, creates a subtle but crucial change in meaning, as by allowing headlines to be both negative and positive at the same time, the authors do not, as written in the introduction, examine whether “negative headlines increase engagement” (p. 3), but the effect of negative words (which may very well appear in a ‘positive’ headline). This distinction is never made explicit in the manuscript; in fact, the wording in the introduction (“negative stimuli”, “negative information”, “negative headlines”, “negative news”) seems to suggest that the authors equate the amount of negative words with the overall negative valence of a piece of news.”

Response: Thanks for raising these important distinctions. We have clarified the terminology and now carefully distinguish between “negative headlines” vs. “negative words” in the introduction. Furthermore, we have added a robustness check by running our analyses on a subset of the data where all headlines that contain both positive and negative words are removed. Headlines with negative words still yield higher click-through rates than headlines with positive words when all headlines that contain both positive and negative words are excluded. These results have been added to Supplement F.5; these demonstrate that our findings are robust.

Comment R3.5: “Consider an extreme example: The certainly positive, made-up headline “Disaster averted: We have conquered war, famine and death” contains only negative (and neutral) words according to the Bing dictionary. This leads to the second point raised above, the domain-specific validity of the dictionary, which is especially relevant given the object of investigation: Upworthy is, by their journalistic self-conception, an outlet for positive news and storytelling and, as such, an unusual choice to study the effects of “negative news” (p. 4). Thus, the authors need to demonstrate with a manual validation that they are indeed able to measure negative sentiment with the Bing dictionary (or any other dictionary, for that matter) in this specific context. (Upworthy may be an ‘outlier’ in another respect as well, in that it had a reputation of being especially ‘click-baity’; as such, users may select news on Upworthy based on different criteria than on, say, more matter-of-fact oriented news outlets).”

Response: Your point is well taken. We added such a validation based on two new user studies in Supplement D. For details, we kindly refer to our response to Comment R3.3 summarizing the

added validation studies.

Comment R3.6: “The unique situation of reviewing two very similar research proposals at the same time provides me with the opportunity to not only review each proposal on its own, but also to look at relative (dis-)advantages. While sharing some of the validation problems outlined above, the ‘other’ submission does explicitly and both theoretically and methodologically address the tenuous connection between negative words and negative sentiment; it also provides several additional robustness checks (e.g., negations of words) and contextualizes the results with the topics and themes of the articles in the dataset.”

Response: Thank you. This is indeed a unique situation but, importantly, both manuscripts arrive at consistent findings. By combining both into a single manuscript, we are confident that we have remedied potential weaknesses and have a collaborative manuscript of overall substantially higher quality. In particular, the collaborative manuscript now contains: (1) a careful wording describing that we examine the effect of negative words; (2) an extensive set of robustness checks (e.g., comparing results with different approaches to negation handling; see Supplement F.2); and (3) a topic model in which we analyze variations in the negativity effect across different news story themes (Supplement H).

Comment R3.7: “1. van Atteveldt, W. & Peng, T.-Q. When communication meets computation: Opportunities, challenges, and pitfalls in computational communication science. Commun. Methods Meas. 12, 81–92 (2018).

2. *Grimmer, J. & Stewart, B. M. Text as Data: The Promise and Pitfalls of Automatic Content Analysis Methods for Political Texts. Polit. Anal. 21, 267–297 (2013).*
3. *Chan, C. et al. Four best practices for measuring news sentiment using ‘off-the-shelf’ dictionaries: a large-scale p-hacking experiment. <https://osf.io/np5wa> (2020) doi:10.31235/osf.io/np5wa.*
4. *Song, H. et al. In validations we trust? The impact of imperfect human annotations as a gold standard on the quality of validation of automated content analysis. Polit. Commun. 1–23 (2020) doi:10.1080/10584609.2020.1723752.”*

Response: Thank you for the references, which we followed closely when performing our user studies for validating the dictionary approach. Where applicable, we also added the references to our manuscript.

Response letter for manuscript:

“Emotions drive online news consumption: Evidence from large-scale field experiments”

Manuscript #2 / Response to Reviewer 1

Comment R1.0: “The authors present an interesting study with a unique dataset. I believe that testing the effects of emotional sentiment in news headlines is important and will be of interest to many readers.

My comments mainly relate to the theoretical framework that was used in this study.”

Response: Thank you for the positive feedback. We proceeded with great care when combining the two papers into a collaborative manuscript. In the revised manuscript, we primarily use the theoretical framework of the other manuscript (“*If it bleeds, it leads: Negative words in headlines elicit greater engagement with news stories*”) and, as such, examine a negativity bias in online news consumption. As a result, some of the comments below regarding the theoretical framework are no longer relevant due to the new focus. Below we provide detailed answers to each comment (and outline improvement actions).

Comment R1.1a: “1. I am not so convinced by the emotion model (Plutchick) that was used as a background for this study. First, I am not sure how far this is still an established model, as it seems to be less frequently used in the scientific literature than other emotion models. Most importantly, I am not sure if I agree with one of the main premises, namely that anger is a positive emotion. I think, traditionally, anger has been conceptualized as a negative emotion. Anger might have positive components or can be beneficial for the person showing this emotion, but I think that the general notion is that anger is a negative emotion. I would therefore rather see the findings when anger is coded as a negative emotion. Or I need a more compelling argument for using anger as a positive emotion.”

Response: We agree that the classification of anger as a positive emotion is contentious. In the original manuscript, we required that anger be a positive emotion to create a balanced sentiment score. We have corrected this in the current manuscript by using the LIWC dictionary for positive and negative emotions, and using the NRC dictionary (which uses Plutchick’s categories) as an exploratory analysis only. By switching to the LIWC dictionary for our sentiment analysis, we eliminate the need for anger to be a positive emotion. That aspect of the theory is no longer

relevant, and has been changed in the manuscript. We continue to use the NRC dictionary for our exploratory analyses of discrete emotions. We included emotion analysis in order to increase the richness of our findings, as prior work has examined the effects of discrete emotions such as anger (Soroka et al. 2015) on news consumption. Furthermore, we conduct our discrete emotion analysis using the Plutchik model because the NRC lexicon is based on the Plutchik model. The NRC lexicon is one of the most comprehensive emotion lexicons today, especially for discrete emotions.

Nevertheless, we saw the need to revise our motivation behind examining emotional words. We no longer motivate our emotion analysis based on the Plutchik model; instead, we motivate the analysis through the prominence of the NRC lexicon (see Mohammad 2016). In a similar vein, we have shifted the focus of the manuscript so that the central focus is around negativity and, thereby, the Plutchik model is no longer relevant to our study to the extent it was previously. To reflect this, our previous section on the Plutchik emotion model is now removed. Altogether, we see the emotion analysis as a secondary analysis that, inspired by prior research (e.g., Chau et al. 2020; Soroka et al. 2015), provides richer insights into the role of discrete emotions.

Comment R1.1b: "Moreover, I am not convinced by Plutchik's dyads. Can the authors elaborate more on why using this model and not another emotion model?"

Related to this, concerning the methodology, what is the advantage of the bipolar emotions, and why were they calculated as mentioned on page 10-11?

Dyads: the reasoning for using the dyads is also not clear to me, what does this add? Why not just investigate the differences between the 8 basic emotions, or just positive vs negative?"

Response: Following your question, we reframed the story of our paper around comparing positive vs. negative sentiment. This presents our main analysis. In addition, we report results for emotional words as a secondary analysis. We discuss our methodological choices (as well the changes during the revision in the following).

- *Choice of the emotion lexicon / emotion model.* It is true that the previous version of our manuscript was based on Plutchik's emotion model. However, the focus of our paper shifted significantly during the revision. Our revised analyses focus on the effect of negative words. We study discrete emotions but this should be regarded as a secondary "exploratory" analysis to offer additional insights for certain discrete emotions. This is inspired by other research where discrete emotions were studied (e.g., Chau et al. 2020; Soroka et al. 2015). We improved our manuscript as follows. In our revised manuscript,

we no longer motivate our analysis based on the Plutchik model. Instead, we motivate our study based on the NRC emotion lexicon which is regarded as one of the most prominent and comprehensive word lists for examining discrete emotions in text (Mohammad, Saif M. "Sentiment analysis: Detecting valence, emotions, and other affectual states from text." *Emotion measurement*. 201-237, 2016; Mohammad, Saif M. "Sentiment analysis: Automatically detecting valence, emotions, and other affectual states from text." *Emotion Measurement*. 323-379. 2021). As part of this, the previous section on "Emotion model" was removed.

- *Bipolar emotions*. We provide an analysis based on the 8 basic emotions in Supplement I. Specifically, we regressed each of the 8 basic emotions on click through rate separately. However, there is a strong linear dependence among the basic emotions, because of which we refrain from adding all basic emotions to the model at the same time. Instead, we added and estimated them via separate models. However, this has the disadvantage that we cannot make statistical inferences that compare basic emotions among one another. To alleviate this problem, we mapped the basic emotions onto bipolar emotions. The bipolar emotions represent the scale of emotions that are least similar to one another. Methodologically, they are relevant because they allow for all 8 emotions to be examined in the same model without making the model rank deficient, considering the unique structure of the NRC emotion lexicon. Hence, we address the linear dependence and can estimate a model where all emotions are included via these bipolar. Reassuringly, the results from the basic emotions support those from the bipolar emotions.
 - *Use of dyads*. We use dyads because prior work has suggested that certain discrete negative emotions such as outrage (Crockett 2017) or contempt/disapproval (Rathje et al. 2021) may be particularly important in online content consumption. We have classified the dyad analyses as secondary analysis that is not part of our main story but should offer richer insights. This has been updated in the manuscript.

Following your questions, we have improved our paper as follows. We now changed the storyline, so that the prime focus is around comparing positive vs negative words. We also report results from the basic emotions in Supplement I but ask for caution when interpreting the estimates. As stated above, the basic emotions are subject to linear dependence (i.e., strong correlation) because of which comparisons between them are not meaningful. Instead, a tailored estimation approach is used as described above.

Comment R1.2: "The authors state in the theory section that"randomized experiments have found that estimates of the influence of online media....can be incorrect by more than 100%". Can the

author further clarify this sentence? What types of estimates, and in which/how many studies has this been found (and which topic)?”

Response: While combining the papers, this sentence was removed from the current draft of the manuscript, and is no longer relevant.

Comment R1.3: “3. The authors argue that “if negative emotions drive news consumption, the same negative emotions are likely to elicit negative emotions among readers.” I think this is somewhat overstated, and not necessarily true. If I read a negative message about a political opponent, I might feel more positive instead of negative. Also the sensational value of negative news can have positive effects on audiences (e.g., entertainment, enjoyment) rather than negative effects (for example true crime series are popular and perceived as entertaining and enjoyable).”

Response: We agree that the statement was misleading and that the present work cannot definitively claim that negative words elicited negative emotions in readers. We removed the corresponding statement. Throughout our manuscript, we have edited and clarified our language, and would like to stress that we are not claiming that emotional words in headlines are corresponding with a news reader’s actual emotion. Your example is well taken -- it is conceivable that an angry headline about an outgroup member would create joyful feelings in a reader. However, our research is ambivalent about what participants’ experienced emotions are, and is solely concerned with examining the type of language that increases news consumption -- we are only analyzing their decision to click on a given headline. This is an important clarification which we have improved in the manuscript in its current Stage 1 state, and are happy to elaborate these caveats in a discussion section at the Stage 2 of the Registered Report.

Comment R1.4: “4. The authors mention that “a better understanding of the emotion effect on news consumption can inform how to curb the proliferation of misinformation.” Maybe the authors can elaborate on this a bit more. How can an understanding of the fact that negative news are clicked on more frequently curb the proliferation of misinformation?”

Response: We agree that elaboration is needed here. Unfortunately discussion and future directions are not permitted in the Stage 1 of Registered Reports. However, we believe that understanding the drivers of news consumption can aid in the potential reduction of fake news in several ways. First, making the public aware of their biases towards negative news may help people self-regulate their media diets more effectively. For example, understanding the drivers of one’s own attention allows for introspection and redirection. Secondly, creating awareness of negativity bias in the news, in addition to other interventions, can be empirically tested as

interventions. We plan to elaborate on this claim further in a discussion section at Stage 2 of the Registered Report. For now, we have removed the above statement from our manuscript.

Comment R1.5: “5. Is the gunning-fox index still used and accepted?”

Response: This is a relevant question. Indeed, the Gunning-Fox index has found application in recent papers examining both online content (e.g., *Zhanfei Lei, Dezhi Yin, Han Zhang (2021) Focus Within or On Others: The Impact of Reviewers' Attentional Focus on Review Helpfulness. Information Systems Research, forthcoming*) and news readability (e.g. *Dougal, Casey, Joseph Engelberg, Diego Garcia, and Christopher Parsons, 2012, Journalists and the stock market, Review of Financial S; Lawrence, Alastair, 2013, Individual investors and financial disclosure, Journal of Accounting & Economics 56, 130-1*). Nevertheless, we acknowledge that there are also other readability indices. Hence, we added a new robustness check (see new Supplement F.3) in which we repeat our analysis based on alternative text complexity scores from the literature (e.g., Flesch, ARI). We still kept the Gunning-Fox index in our main analysis due to its widespread use, straightforward interpretation, and robust performance. Overall, our main findings remain robust. Of note, we refrained from using scores based on artificial intelligence, since these have been typically calibrated on texts from a different domain (e.g., books) and thus cannot accurately infer the complexity due to a domain shift which would arise from short headlines, thus implying biased results. Instead, we used alternative approaches that are used as robustness checks in other research (e.g., *Lehavy, Li, & Merkley, 2011 The effect of annual report readability on analyst following and the properties of their earnings forecasts. The Accounting Review, 86, 1087-1115; Hu, Chen, & Lee, 2017, The effect of user-controllable filters on the prediction of online hotel reviews. Information & Management, 54, 728-744*).

Comment R1.6: “6. Question about the CTR: are these somehow controlled for the same persons being part of several RTC? Are the data nested? Do we know how many users did participate in these trials?”

Response: Excellent questions.

- (1) The Upworthy Research Archive data is completely aggregated, and there is no way to look at patterns of engagement at the individual level for privacy. Of note, all randomized controlled trials were conducted separately, so that bias is avoided.
- (2) The data are nested at story level (i.e., each story has a separate randomized controlled trial). In the revised manuscript, we use a multilevel model to account for the between-story heterogeneity. We clarified the wording and now use the term “multilevel model” more explicitly.
- (3) We know how many people were assigned to each variation of an A/B test. This denoted

by the variable $impressions_{ij}$ based on which we calculate the click-through rate. On average, each randomized controlled trial involves approximately 16,670 users (participants) that were then assigned to the different variations.

To improve our manuscript, all of the aforementioned points have been clarified in the manuscript. We also added further descriptions and a graphic to explain the structure of the data set.

Comment R1.7: “7. Does the sentiment analysis take negations into account, as in the example (“not amused” indicating irritation rather than amusement)? Do we know how accurately the sentiment analysis reflects the emotionality of the headlines? Was this compared to manual coding of a subsample?”

Response: We improved our manuscript in two ways: (1) Yes, we account for negations. For example, the phrase “not fun” would be coded as negative rather than positive due to the negation. Based on your feedback, we changed our methodology and now include negation handling in all analyses. We clarified the methodological details in the revised “Methods” section. We also experimented with an alternative approach to negation handling. The results for this are reported in a new Supplement F.2 but are qualitatively identical to those from our main analysis. This adds to the robustness of our results. (2) We additionally performed two user studies as a validation. The user studies confirm that our sentiment analysis reflects the emotionality of headlines as perceived by users. The new analyses are reported in a new Supplement D.

To facilitate review, we copy a detailed description of the two actions from above:

- **Negation handling.** We included negation handling into our text mining framework. This is now—by default—part of the main analysis. Now, we reverse the sentiment direction when negation is present. For example, the phrase “not fun” would be coded as negative rather than positive due to the negation. More formally, the following procedure is used. First, the text is scanned negation terms using a predefined list, and then all emotional words in the neighborhood are counted as belonging to the opposite sentiment (or emotion). In our analysis, the neighborhood is set to 5 words before and 2 words after the negation. Details are in the Methods.

We also added a new analysis comparing the above approach to negation handling based on an alternative neighborhood (i.e., the so-called negation scope) to confirm the robustness of our findings. Both analyses led to qualitatively identical results (see new Supplement F.2).

- **Validation based on user study.** To ensure that our dictionary analysis captured emotionality in headlines, we performed two new user studies to validate the accuracy of our dictionary approach (see new Supplement D). Study 1 validates that user’s judgments

of emotionality in headlines correlate with the number of negative and/or positive words each headline contains. We found user judgments of sentiment and our sentiment scores were significantly positively correlated at $r_s = .36$ ($p < .001$). These findings validate that our dictionary approach captures significant variation in the perception of emotions in headlines from users. Study 2 validates that users ($k = 10$ as in Song et al. 2020) perceive headlines with discrete emotional words (such as angry words) as reflecting that emotion. We found that the overall correlation between NRC dictionary scores and user judgments for the 8 discrete emotions was positive and statistically significant ($r_s = 0.13$; $p < 0.001$). Detailed results for each dimension of the NRC emotion lexicon are in the new Supplement D. This validates that our dictionary approach captures the perceptions of emotions from users.

Manuscript #2 / Response to Reviewer 2

Comment R2.0: "This paper uses a rich dataset of news article RCT dataset to study the relationship between emotions and news engagement. Use of the Upworthy RCT dataset allows the authors to identify the effect of emotions embedded in the news articles while accounting for other confounding factors such as the body of the article. The authors use Plutchik's emotion model, which identifies a comprehensive set of basic emotions as well as more complex ones that are derived from them."

While I found aspects of the paper interesting, I have some concerns."

Response: Thank you for the positive assessment. We carefully followed your suggestions below and, based on them, improved our work. We think your feedback was very helpful in bringing our manuscript to a substantially higher quality.

Comment R2.1: First, I appreciated that the authors provided example headlines for the reader to inspect (table 1). However, inspecting it, I am more concerned that the dictionary based approach might not be corresponding to human perception of emotions. For instance, the 4 sentences are positive, neutral, negative, and neutral, if I didn't miscount the colored words. But they look rather comparable. There are some recent studies that question the use of emotion dictionaries (eg. Kross, Ethan, Philippe Verduyn, Margaret Boyer, Brittany Drake, Izzy Gainsburg, Brian Vickers, Oscar Ybarra, and John Jonides. "Does counting emotion words on online social networks provide a window into people's subjective experience of emotion? A case study on Facebook"). It would be great if the authors can perform some sort of validation test here, perhaps getting human

judgment on the emotions conveyed by the headlines, perhaps gathered through crowdsourcing platforms etc.

Response: This is a relevant question and we addressed it with great care:

- (1) **Revised example.** Upon careful inspection, we found that the selected examples are not helpful for potential readers. Instead, we carefully revised the examples (see revised Table 1 in the main manuscript, and copied below.).

#	Headline Variation	CTR
1	If The Numbers 4 And 20 Mean Something To You, You're Gonna Want To Hear This Shit	0.94%
2	What He Has To Say About Pot Is Going To Make Both Sides Angry , But Here Goes	0.79%
3	Lots Of Things In Life Have Both A Benefit And A Harm . So Why Do We Only Obsess About This One?	0.60%
4	He Explains Why The Question 'What Are You Smoking' Is Actually Kind Of Important .	0.58%
1	IMAGINE: You're Raped At Your Job And Your Boss Intentionally Tries To Shut You Up	0.92%
2	12 Minutes. If You Support Our Troops, Sacrifice At Least That Much For Them.	0.21%
1	Spoofers Set Up A Fake Agency To Show How Ridiculous Some People Are When It Comes To Immigrants	0.65%
2	Something's Been Missing From Our Favorite Superhero Stories, And It Makes Reality Seem Kinda Silly	0.56%
3	Some Comic Book Lovers Might Need To Check Their Politics When They See What These Guys Have In Mind	0.53%
4	A New Agency Wants To Get Rid Of All Our Favorite Superheroes. I Laughed When I Saw Why.	0.41%
1	I Knew Which One She'd Pick, But It Still Crushed Me	1.10%
2	First She Points To The Pretty Child. Then To The Ugly Child. Then My Heart Breaks.	0.85%
3	1 Little Girl, 5 Cartoons And 1 Heartbreaking Answer	0.83%
4	What She Says About These Cartoons, Says Something Incredibly Troubling About The World We Live In	0.66%

Legend: **Positive** , **Negative**

- (2) **Validation based on user study.** We performed two user studies to validate the accuracy of our dictionary approach (see new Supplement D). Study 1 validates that users ($k = 10$) perceive headlines with negative words as being more negative. Study 2 validates that users ($k = 10$) perceive headlines with discrete emotional words (such as angry words) as reflecting that emotion. This confirms that our dictionary approach captures the actual perceptions of emotions.

- (3) **Differences to Kross et al. 2020.** The present work differs in several important ways from the critique presented in the Kross et al. 2020 citation suggested. First, we are not

claiming that emotional words in headlines are corresponding with a news reader's actual emotion. For example, it is conceivable that an angry headline about an outgroup member would create joyful feelings in a reader. However, our research is ambivalent about what participants' experienced emotions are, and is solely concerned with examining the type of language that increases news consumption. Secondly, the paper cited above refers to interpreting an individual's emotions based on participant's own social media posts. However, Upworthy.com was not a social media website, and we are not analyzing readers' comments or even their sharing -- we are only analyzing their decision to click on a given headline. This is an important clarification which we have improved in the manuscript in its current Stage 1 state, and are happy to include these caveats in a discussion section at the Stage 2 of the Registered Report.

- (4) **Additional materials on emotion lexicons.** We have also added a table listing the most common negative and positive words (see new Supplement C). We also list a similar table with the most common emotional words.

Comment R2.2: "Second concern I have is the number of models/effects tested. While it is interesting that the emotion model allowed the researchers to delve deeper to provide a deeper look, it also led them to testing various hypotheses. It would be best if the authors can perform multiple hypothesis testing corrections."

Response: We conducted a Bonferroni correction for multiple hypothesis testing. The results remain robust. Details are stated in the main paper.

Comment R2.3: "Third, it would be great to get more insights about the RCTs that created the data used in this paper. Authors mention that there are multiple packages with identical headlines for instance. Not sure why. I would also be curious to learn more about whether the RCTs were generally varying some other factor (say, sarcasm). These details maybe don't need to be in the main manuscript but more insights about the data would help."

Response: Thank you for giving us the opportunity to expand our elaborations on the Upworthy archive and, specifically, the RCT design.

- (1) *Terminology.* The term "package" was internally used by Upworthy and is used in the Upworthy archive when referring to the headlines in a single randomized controlled trial (RCT). To aid readability, we replaced the term "package" by "RCT". A RCT involves around ~4 different headline variants for the *same* news story that were evaluated

- experimentally. We clarified our wording throughout the manuscript.
- (2) *Preprocessing*. Based on your question, we saw the need to clarify the wording regarding how the data were handled. We preprocessed the data from RCTs as follows. The original dataset contains experiments where identical headline variations were stored multiple times in the Upworthy archive. For instance, the same headline text was stored under two different database identifiers. We combined the corresponding clicks / impressions into a single variable.
 - (3) Other factors (images). The reason that some tests had identical headlines were that there were also tests of other features occurring concurrently (image testing). Unfortunately, the Upworthy research archive does not make image data available to researchers. We have added more details about the features of each headline that can vary. To ensure the robustness of our findings, we performed additional checks (see new Supplement F.6). Here, the results do not change when only headlines with no image test are included in the analysis.
 - (4) Additional insights. We added additional insights on the RCTs (here, the suggestions from the whole review team have been very helpful). For instance, we added more descriptives such as word frequencies in the RCTs etc.

Comment R2.4: "The sentiment model: the authors had mentioned in their discussion of related work that some studies find stronger engagement for content with stronger sentiment (e.g. very positive and very negative). I am curious why they went with a linear model for sentiment at the end (formula 3). There are already various models tested so perhaps it is not ideal to add but curious about the decision process that led to excluding this as an alternative explanation to test."

Response: This is indeed a very relevant thought and we improved our manuscript as follows.

We changed the model specification: we replaced the *Sentiment* variable (which was the difference between positive and negative) by two separate variables *Positive* and *Negative*. This allows us to control for a "very positive" and/or a "very negative" tone, and is thus key to identify a negativity bias. For this, we updated all equations, as well as all results related to positive/negative emotions.

Comment R2.5: "Topic modeling: The topic model is not evaluated. As such, I find it hard to take the results at face value. Topic models commonly fail in document assignment despite top-k words looking good (table-6 has decent top-10 words). I would encourage the authors to validate this model before using. Here, again, they can rely on human judgment on a subset of headlines. The authors might also consider using the body of the article in performing this topic analysis."

Response: We agree that validation is important for topic modeling. We performed an additional user study to validate our topic models (see revised Supplement H). For this, we took inspiration from (Brady et al., 2017) and followed the approach in (Chang et al., 2009), which is considered best-practice for validating topic models. The objective behind our user studies is to validate that participants were better than chance at categorizing headlines as belonging to a certain topic in accordance with our topic models (“topic intrusion”). Participants ($k = 10$) were significantly above chance for correctly assigning each headline to its correct topic. The new and extensive analysis strengthens the validity of our findings. Taken together, the user study confirms that the model combines news headlines into meaningful topics.

We also considered conducting a topic model on the body of the text, yet this was not feasible because we do not have access to the full text of the articles.

Comment R2.6: “Minor note: I am not an emotions researcher. I found it surprising that anger was classified under positive sentiment. Is this widely accepted or only in the Plutchik model?”

Response: By switching to the LIWC dictionary, we eliminate the need for anger to be a positive emotion. That aspect of the theory is no longer relevant, and has been changed in the manuscript.

Manuscript #2 / Responses to Reviewer 3

Comment R3.0: “The proposed research, ‘Emotions drive online news consumption: Evidence from large-scale field experiments,’ seeks to investigate the effect of emotions on the click-through rate of news headlines. To that end, the authors propose the use of the full Upworthy Research Archive and present their planned methodology using a small subset of the data. The research question is certainly relevant for a broad, multidisciplinary audience and I concur that the dataset might provide compelling answers, some limitations (see below) notwithstanding.

This review presents somewhat of a novelty to me, as I was asked by the journal to review two similar stage 1 registered reports by different authors. Both proposals focus on the effects of emotion, especially negative emotion, on click-through rates. Both proposals use the Upworthy Research Archive. Both proposals share the main preliminary conclusion that negativity indeed seems to causally increase news click-through rates.”

Response: Thank you for the positive feedback, and for seeing that our research is relevant for a broad, multidisciplinary audience. Importantly, while both analyses vary in their methodology, both share the same main findings. It was indeed an interesting process to combine the two papers

into a collaborative manuscript, and, during this, we followed your suggestions closely.

Comment R3.1a: “Both proposals also share their biggest weakness in that they rely on (different) off-the-shelf sentiment dictionaries, but fail to (re-)validate them for the specific context they are used in. Validation, of course, is absolutely crucial in the context of automated text analysis: “The validity of a method or tool is dependent on the context in which it is used, so even if a researcher uses an existing off-the-shelf tool with published validity results it is vital to show how well it performs in a specific domain and on a specific task.” 1–3 A manual validation of the used dictionary on the dataset the analyses are performed on—Song et al. recommend 1% of the data⁴—should thus demonstrate that the bag-of-words and additivity assumptions (more negative than positive words = negative sentiment, and the bigger the difference, the more negative) holds true and that the dictionary can be used on the thematic domain of the data.”

Response: Here, we repeat our improvement actions that we also used in our response to manuscript #1.

Thank you for this helpful suggestion! We agree that our original analyses could not adequately answer the question of whether headlines with more negative words were perceived as more negative. We thus followed best practices (e.g., Song et al. 2018) and performed two additional user studies as validation. The objective behind this was to ensure that our dictionary analysis captured emotionality in headlines. We included $k = 10$ raters, and examined 2.3% of our data in the validation, i.e., 2x the amount recommended by Song et al (2018). We added the two user studies in a new Supplement D. Study 1 validates that users perceive headlines with negative words as more negative. Study 2 validates that users perceive headlines with discrete emotional words (such as angry words) as reflecting that emotion. Across both, we compared the accuracy of the dictionary approach against the judgments of human raters. Overall, we find that our dictionary approach is successful in capturing the actual perceptions of emotions in the thematic domain of the data.

Furthermore, the correlation between user ratings of sentiment and computer calculated sentiment speaks to the additivity assumption. (1) A headline with two negative words is perceived as more negative than a headline with only one negative word, and so on. (2) A headline with two negative words is perceived as more negative than a headline with only one negative word, and so on. (3) A text with more negative than positive words is perceived as having a negative sentiment. (4) A text with more positive than negative words is perceived as having a positive sentiment. Altogether, the computer calculated sentiment has a strong and statistically significant association with sentiment ratings by users, thus implying that our dictionary approach captures the sentiment well.

As an important note, we asked participants whether they thought the headline they were reading was more negative or positive, not whether the headline made them feel more negative or positive. This is an important distinction, as most studies on negativity bias to this point have focused on the subjective experience of participants (i.e., Kross et al. 2019). Our study, on the other hand, is ambivalent towards the reader's emotional experience. We are only making claims based on participants' quantifiable behaviour (i.e., news consumption).

When combining the two manuscripts, we found that each relied upon a different sentiment lexicon. We thus incorporated an additional robustness check (see revised Supplement F.1) where we compare the results against alternative sentiment lexicons (LIWC, NRC, and SentiStrength). This should further demonstrate that, regardless of the lexicon, the conclusions remain robust.

Comment R3.1b: "Regarding the first point, the additivity assumption is explicitly used in the calculation of overall negative vs. positive sentiment as well as in the calculations of bipolar emotions. The validity of this assumption is especially crucial in the context of the NRC dictionary where the same words can load onto both 'negative' and 'positive' emotions (which leads to some peculiarities, such as the word 'suicide' being coded as overall neutral, whereas 'giant' is an overall negative word, see p. 36; Table 3 also suggests that 'lying' is considered to be an emotional word in the present tense, but not in the past tense)."

Response: We now use the LIWC dictionary and model negative and positive sentiment separately, eliminating the additivity assumption. Furthermore, some of the peculiarities of the NRC dictionary are eliminated by switching to LIWC. The manuscript has been updated to reflect this change. For clarity, we have included a table of the most common positive and negative words from our sample in Supplement B. Furthermore, we conducted a new validation study that validates that users ($k = 10$) perceive headlines with negative words as being more negative. User results and the dictionary-based approach were significantly correlated with one another, suggesting that a greater number of negative (positive) words in headlines was associated with an increased perception of negativity (positivity).

Comment R3.2: "Second, the domain-specific validity of the dictionary is especially relevant given the object of investigation: Upworthy is, by their journalistic self-conception, an outlet for positive news and storytelling and, as such, an unusual choice to study the effects of "negative emotions" (p. 6). Thus, the authors need to demonstrate that the NRC is indeed able to adequately measure the eight emotions and negative/positive sentiment in the context of Upworthy. For example, to use the examples provided by the authors, they need to show that "Some College Kids Figured

Out How To Cut Energy Companies Out Of Our Lives. They Are Not Amused.” is indeed perceived to be more positive than “Some Young Punks Just Punked A Giant Corporation Out Of The Easy Money It Was Bilking Off You”, which in turn is perceived to be more positive than “Note To Giant Corporation: Don’t Mess With Young Punks Who Have More Brains Than You Have Money”, with each headline leading to the appraisal of the associated emotions. This validation is especially relevant in the context of comparatively short headlines where a single word might substantially alter the coded sentiment.”

Response: We have switched to the LIWC dictionary, which is better established in the social sciences. We agree that the negativity bias is surprising in the Upworthy dataset, due to its self-proclaimed positivity. We believe, however, that this makes the case for negativity bias *stronger*, as our results suggest that even positive news sites rely on negative words and sentiment to drive engagement. This point has been added to the manuscript. In addition, we included two user studies as validation (see new Supplement D). The additional validation is summarized in our response to Comment R3.1a.

Comment R3.3: “One additional limitation of the dataset is that Upworthy had a reputation of being especially ‘click-baity’; users were thus likely to expect emotional wording in the headlines, which might affect the representativeness of Upworthy user behavior for all online news.”

Response: We agree that Upworthy’s “click-bait” tendencies may influence clicking behavior. However, due to the extremely high readership of upworthy during the height of its popularity, in addition to it being a free news source, we believe the Upworthy Research Archive is among the most (if not the most) representative sets of news consumption data that is freely available to researchers. We will discuss this further. However, discussions are not part of Stage 1 in Registered Reports but we will elaborate on this further in Stage 2 of the Registered Report.

Comment R3.4: “The unique situation of reviewing two very similar research proposals at the same time provides me with the opportunity to not only review each proposal on its own, but also to look at relative (dis-)advantages. While sharing some of the validation problems outlined above, the project at hand provides a more granular approach, contains additional robustness checks (e.g., negations of words) and contextualizes the results with the topics and themes of the articles in the dataset (however, topic models should be manually validated as well, for example with a word intrusion test). Thus, should the editors decide to move forward with only one of the submitted projects, I would suggest it to be this one. However, the validity of the automatic emotion coding, as outlined above, needs to be established before an acceptance in principle.”

Response: Thank you for the positive feedback. We followed your suggestions:

- (1) **Validation of dictionary approach.** We added two user studies to validate the accuracy of the dictionary approach. For details, see our response to comment R3.1a. The validation is added as a new Supplement D.
- (2) **Granular insights.** In combining the manuscript, we framed our main story around the negativity bias. We further kept the granular analysis across different emotions as before (but decided to label it as a secondary “exploratory” analysis).
- (3) **Robustness checks.** We kept all robustness checks as before. Based on feedback from the reviewers, we added additional ones (e.g., by including different readability scores).
- (4) **Validation of topic model.** We agree that validation is important for topic modeling. We took inspiration from (Brady et al., 2017) and performed an additional user study to validate our topic models (see revised Supplement H). For this, we followed the approach in (Chang et al., 2009), which is considered best-practice for validating topic models. The objective behind the user study is to validate that participants were better than chance at categorizing headlines as belonging to a certain topic in accordance with our topic models (“topic intrusion”). The new and extensive analysis strengthens the validity of our findings. The user study confirms that the topic model combines news headlines into meaningful topics.

We considered the use of a word intrusion test but eventually discarded this. The reason is that we fitted our topic model to only $k = 7$ topics, which returns comparatively broad topics that should reflect sub-categories in modern news websites (e.g., “Entertainment”, “Government and Economy”, etc.). Hence, our topic model does not combine news headlines that belong to the news coverage of a specific event (e.g., all news stories around a new presidential election, or all news stories around a specific economic downturn), which one would obtain using a granular categorization. Instead, our topic model identifies broader thematic areas (e.g., “Government and Economy”). Therefore, it is reasonable to expect a large diversity among the news headlines within the same topic. Because of this, it will be difficult for users to identify intruding words since the word lists (similar to the topics) should be comparatively broad. Hence, we rather opted to report example headlines per topic to show that the headlines within each topic form meaningful entities. This is further justified as we only analyze the between-topic heterogeneity in our downstream regression analysis (and not the within-topic similarly). Specifically, we are interested in a broader thematic categorization where emotion effects differ and in a way that such insights can then be leveraged in a downstream regression analysis. This is also discussed in our revised manuscript. Our choice of testing the topic categorization rather than the word categorization is also in line with other social science research (e.g., Brady et al., 2017). To improve our manuscript further, we additionally extended our manuscript and added a table with example headlines per topic to aid better interpretation by readers (see revised Supplement H.2). For transparency, we report the words for each

topic to allow for a manual inspection (see Supplement H).

We hope you also find merit in combining the two manuscripts into a collaborative work that is of even higher quality.

Comment R3.5: "1. van Atteveldt, W. & Peng, T.-Q. When communication meets computation: Opportunities, challenges, and pitfalls in computational communication science. Commun. Methods Meas. 12, 81–92 (2018).

2. *Grimmer, J. & Stewart, B. M. Text as Data: The Promise and Pitfalls of Automatic Content Analysis Methods for Political Texts. Polit. Anal. 21, 267–297 (2013).*
3. *Chan, C. et al. Four best practices for measuring news sentiment using 'off-the-shelf' dictionaries: a large-scale p-hacking experiment. <https://osf.io/np5wa> (2020) doi:10.31235/osf.io/np5wa.*
4. *Song, H. et al. In validations we trust? The impact of imperfect human annotations as a gold standard on the quality of validation of automated content analysis. Polit. Commun. 1–23 (2020) doi:10.1080/10584609.2020.1723752."*

Response: Thank you for the references, which we followed closely when performing our user studies for validating the dictionary approach. Where applicable, we also added the references to our manuscript.

Decision Letter, first revision:

16th August 2021

Dear Stefan,

RE: "Negativity drives online news consumption: Evidence from large-scale field experiments"

Thank you for submitting your revised manuscript, and for all your work to combine author teams.

Although your manuscript has been revised in response to reviewer and editor comments, it does not fully comply with our editorial policies and formatting requirements for Stage 1 Registered Reports. For example, the pre-registered hypotheses do not currently appear in the main text, there is no required Design Table (this is key, as we require Stage 1 Registered Reports to outline how hypotheses will be tested and how results will be interpreted), and power calculations are not included. While the Discussion section may be outlined in the response to reviewers document, it should not appear in the Stage 1 manuscript. Additionally, the pilot data should be clearly distinguished from the main analyses. Rather than updating the numbers once the full dataset is available, the preliminary results should be separately presented as pilot results, with the full results to follow in the Stage 2 manuscript. Please also ensure that all pilot and other statistical results are fully reported, including effect sizes/coefficients, p-values,

and confidence intervals.

Before we can send the manuscript back to our reviewers, we ask that you revise it to ensure that it complies fully with our policies and is formatted according to our requirements. I have attached another copy of our template document for Stage 1 Registered Reports, and we ask that you adhere as closely as possible to this template. More information is also available on our website: <https://www.nature.com/nathumbehav/registeredreports>. If you are uncertain as to how to address any of the points in the template, please don't hesitate to contact me.

[REDACTED]

Thank you in advance for attending to these requests and I look forward to receiving your revised manuscript.

Sincerely,
Aisha

Aisha Bradshaw, PhD
Senior Editor
Nature Human Behaviour

Decision Letter, second revision:

23rd September 2021

Dear Stefan,

RE: "Negativity drives online news consumption: Evidence from large-scale field experiments"

Thank you for submitting your revised Stage 1 Registered Report, and for all of your work to address our formatting requests in addition to reviewer comments.

Although your manuscript now largely complies with our editorial policies and formatting requirements for Stage 1 Registered Reports, there are two key issues that remain outstanding. Because these are aspects of the manuscript that we feel reviewers must be able to comment on as part of the peer review process, we ask that you further revise before we send your manuscript back to reviewers.

In particular, we ask that you address the following:

1.) For Registered Reports, the power analysis must be carried out based on the minimum theoretically meaningful effect sizes. That is, your pilot effect sizes should not be used as the basis for determining

effect sizes for power analyses.

2.) Your analysis plan (and the design tables) do not include mention of the methods you will use to interpret null results. If a result is null, it cannot be interpreted unless appropriate statistical analyses are used to enable interpretation.

You can either choose to use Bayes Factors or Equivalence Testing (or both).

This article provides a tutorial on using BF's to interpret null results:

<https://www.frontiersin.org/articles/10.3389/fpsyg.2014.00781/full>

This article provides a tutorial on using equivalence testing for null results:

<https://journals.sagepub.com/doi/10.1177/2515245918770963>

Regardless of your choice, please do make sure that you specify the necessary information to carry out these analyses.

In addition to these core points, you may wish to retain the main results of your pilot study in the main text, to facilitate reader access to these results. However, if you prefer to keep all pilot results in the Supplementary Information, this is not a strict requirement for re-review.

In sum, before we can send the manuscript back to our reviewers, we ask that you revise it to address these two outstanding editorial requirements. More information on these points can be found in the attached template document, or on our website:

<https://www.nature.com/nathumbehav/registeredreports>. If you have any questions about these requested edits, please don't hesitate to contact me.

[REDACTED]

Thank you in advance for attending to these requests and I look forward to receiving your revised manuscript.

Sincerely,
Aisha

Aisha Bradshaw, PhD
Senior Editor
Nature Human Behaviour

Decision Letter, third revision:

10th December 2021

Dear Stefan,

Thank you once again for your revised manuscript, entitled "Negativity drives online news consumption: Evidence from large-scale field experiments," and for your patience during the peer review process.

Your manuscript has now been evaluated by the same 3 reviewers who saw the original version of your work, and their comments are included at the end of this letter. Although the reviewers find your protocol to be of interest, they also raise some important concerns. We remain very interested in the possibility of proceeding further with your submission in *Nature Human Behaviour*, but would like to consider your response to these concerns in the form of a revised manuscript before we make a decision on in principle acceptance and Stage 2 submission.

You will see from their comments that Reviewers 1 and 3 are now satisfied with your protocol. However, Reviewer 2 highlights concerns about the discrete emotion analyses, including about the validation evidence and the extent to which these analyses are well-motivated theoretically. In your revision, we ask that you address these concerns, both through additional explanation of your approach and through alterations to your design or provision of additional evidence as needed.

Editorially, we also note that you currently do not plan to interpret any null results that may arise. This option is consistent with our policy for Registered Reports. However, in order to reduce the risk of an inconclusive study, we encourage you to consider including a plan that would allow for appropriate interpretation of null results (e.g. through the inclusion of Bayes Factors). Stage 1 acceptance in principle is not contingent on following this recommendation, but we do feel that doing so could potentially provide more insight at Stage 2.

In addition to these reviewer points, we also ask that you address some outstanding formatting and editorial issues in order to ensure that your manuscript fully complies with our requirements for Stage 1 Registered Reports. By addressing these points now, we anticipate that time can be saved at future stages of the process. In particular, please address the following:

1. Please ensure that all pilot results are reported in full, including coefficients/effect sizes, p-values, and confidence intervals. For example, the statement on pg. 42 "We found a negative and statistically significant direct effect of moralized language on CTR and negative and statistically significant effects for the interactions between the proportion of moral words and the proportion of positive/negative words" should be supported by fully reported statistical results. In cases where reporting a large number of results in text may negatively affect readability, please instead include a reference to a specific table where full results can be found. Please note that tables reporting these statistical results should include full statistical information, including exact p-values (i.e. asterisks should not be used to denote significance).

2. Please remove the current pg. 3, which explains the arrangements made with Upworthy for

completing a Registered Report. Instead, you can explain in the Methods under what condition the data were made available by Upworthy (i.e., on condition that you will publish a Registered Report).

3. All novelty/priority claims must be removed. For instance, please revise the statement on pg. 5, “we present novel findings” to remove the word “novel”

4. In the Ethics Information section (pg. 9), please indicate whether participants in the user validation study provided informed consent and provide information on participant compensation. Also note that currently, the ethics information section appears twice (once at the start of the Methods and once at the end of the paper on pg. 26). This information only needs to be included at the start of the Methods section.

5. Please include an Acknowledgments section, following the guidance in the attached Template document.

6. Please note that ultimately, your title will need to be revised to follow our formatting requirements. The format of title: subtitle is not permitted, and no punctuation may be used except commas in lists.

In sum, we invite you to revise your Stage 1 Registered Report taking into account reviewer and editor comments. Please highlight all changes in the manuscript text file.

* Include a “Response to reviewers” document detailing, point-by-point, how you addressed each referee comment. If no action was taken to address a point, you must provide a compelling argument. This response will be sent back to the reviewers along with the revised manuscript.

* Ensure that you use our template for Stage 1 Registered Reports to prepare your revised manuscript: https://www.nature.com/documents/NHB_Template_RR_Stage1.docx. Failure to ensure that your revised Stage 1 submission meets our requirements as specified in the template will result in your submission being returned to you, which will delay its consideration.

* In your cover letter, please include the following information:

--An anticipated timeline for completing the study if your Stage 1 submission is accepted in principle.

--A statement confirming that you agree to share your raw data, any digital study materials, computer code (if relevant), and laboratory log for all eventually published results.

--A statement confirming that, following Stage 1 in principle acceptance, you agree to register your approved protocol on the Open Science Framework (<https://osf.io/>) or other recognised repository, either publicly or under private embargo, until submission of the Stage 2 manuscript.

--A statement confirming that if you later withdraw your paper, you agree to the Journal publishing a short summary of the pre-registered study under a section Withdrawn Registrations.

[REDACTED]

We hope to receive your revised manuscript within three months. If you cannot send it within this time, please let us know. We will be happy to consider your revision so long as nothing similar has been accepted for publication at Nature Human Behaviour or published elsewhere.

Nature Human Behaviour is committed to improving transparency in authorship. As part of our efforts in this direction, we are now requesting that all authors identified as 'corresponding author' on published papers create and link their Open Researcher and Contributor Identifier (ORCID) with their account on the Manuscript Tracking System (MTS), prior to acceptance. ORCID helps the scientific community achieve unambiguous attribution of all scholarly contributions. You can create and link your ORCID from the home page of the MTS by clicking on 'Modify my Springer Nature account'. For more information please visit www.springernature.com/orcid.

Sincerely,
Aisha

Aisha Bradshaw, PhD
Senior Editor
Nature Human Behaviour

Reviewer expertise:

Reviewer #1: communication, emotion

Reviewer #2: computational social science, Registered Reports

Reviewer #3: online communication

Reviewers' Comments:

Reviewer #1:
Remarks to the Author:

I complement the authors to join forces in this project. The authors have addressed all my concerns and suggestions, and I therefore recommend to accept the registered report. I believe that this will be a very interesting and relevant paper, and I am looking forward to seeing the final results.

Reviewer #2:

Remarks to the Author:

The new manuscript that combines the two efforts is a significant improvement over both papers. I commend the authors for taking on the revision of their work and creating a coherent story based on two originally separate efforts. The writing is also significantly improved—essential details are clarified, and the paper's structure is much better and easier to follow. I also appreciate that the authors validated both dictionary and topic modeling approaches.

However, I still have some concerns. These concerns are mostly related to the study of discrete emotions. The authors do not sufficiently motivate why bipolar emotion pairs and emotional dyads are valuable to investigate. As far as I understand, the biggest reason for analyzing the bipolar pairs is to limit the search space (compared to examining all emotions). Is that the same for embedded emotional dyads? Without past work that helps us reason about the theoretical basis of why it is essential to look at these measures, I am still not sure exactly what kind of generalizable knowledge we will gain at the end. This worry is further strengthened due to the very weak correlation the authors find for emotions and that not being significant some emotions.

The other concern I have is about the result of validation efforts. As I mentioned above, the correlation for discrete emotions is very weak. The authors also only note that the human judges are better than random chance for the topic intrusion test but not by how much. Better than random is a very weak baseline. How much better? How does that vary across topics? As I had mentioned in my original review, topic models are generally good at prevalent topics but can fail at rarer ones. I can imagine you can get better than random performance on average while having a very poor performance for many rarer topics. Overall, while there are validation efforts, it is unclear whether the bar of "good enough" has been set properly.

A few other points are listed below:

Dictionary words: Some of the words in C.2 make me more worried. The top word for anger is words, the second most important for disgust is boy, top word for fear is watch.

Dictionary validation: (section D)

You say: "The number of headlines ($N = 213$) and the number of raters were chosen based on best practices, laid out explicitly in [60]." Can you please write down your procedure explicitly? The readers shouldn't need to go to a different paper to know about your sampling design.

You say you use spearman correlation but do not explicitly mention with what. Is it the average of 8 human judges' scores or median? Or did you look at each human judge score separately? I am hoping that it is not the latter, but I can't find a reference to this in the paper. These aggregation functions would have different implications, so please clarify.

Correlation of human judgment and dictionary approach: 0.3 is a weak correlation. Please note this in your writing.

For discrete emotions, the correlation is even weaker. This, combined with the dictionary results from C2 and lack of theoretical grounding and interpretation of the related findings, makes me think this part of the paper needs to be strengthened or removed. For instance, given the weak correlation, did you look at the cases where human judgment disagrees with to see if there are some important patterns? You can also see that the correlation is not significant for some emotions.

Reviewer #3:
Remarks to the Author:

I am impressed by the thorough revision and the 'merged' effort to combine two interesting proposals into one study that certainly will provide interesting insights. As my main concern of lacking validation of the entertainment dictionaries has been rigorously addressed by the inclusion of several validation user studies, I believe the study proposal is now ready for stage 2.

Author Rebuttal, fourth revision

Response letter NATHUMBEHAV-200611432D: “Negativity drives online news consumption: Evidence from large-scale field experiments”

Dear Dr. Bradshaw & Reviewers,

Thank you for the opportunity to revise and resubmit our paper “*Negativity drives online news consumption: Evidence from large-scale field experiments.*” We appreciate the time and attention given to our manuscript. Please find below our responses to your comments.

Response to Reviewer #1

Comment R1.1: “I complement the authors to join forces in this project. The authors have addressed all my concerns and suggestions, and I therefore recommend to accept the registered report. I believe that this will be a very interesting and relevant paper, and I am looking forward to seeing the final results.”

Response: Thank you for the kind words!

Response to Reviewer #2

Comment R2.1: “The new manuscript that combines the two efforts is a significant improvement over both papers. I commend the authors for taking on the revision of their work and creating a coherent story based on two originally separate efforts. The writing is also significantly improved—essential details are clarified, and the paper’s structure is much better and easier to follow. I also appreciate that the authors validated both dictionary and topic modeling approaches.”

Response: Thank you for the positive feedback!

Comment R2.2: “However, I still have some concerns. These concerns are mostly related to the study of discrete emotions. The authors do not sufficiently motivate why bipolar emotion pairs and emotional dyads are valuable to investigate. As far as I understand, the biggest reason for analyzing the bipolar pairs is to limit the search space (compared to examining all emotions). Is that the same for embedded emotional dyads? Without past work that helps us reason about the theoretical basis of why it is essential to look at these measures, I am still not sure exactly what kind of generalizable knowledge we will gain at the end. This worry is further strengthened due to the very weak correlation the authors find for emotions and that not being significant some emotions.”

Response: Thank you for raising this relevant concern. In order to address this, we have reorganized the manuscript in the following ways.

- First, we have narrowed our focus of the emotion analysis to only include discrete emotions (e.g., anger, fear, etc.). In particular, we removed several constructs that we had in the previous version of the manuscript (e.g., bipolar pairs, emotional dyads). After careful reflection, we found that the theoretical foundation is not sufficiently strong and that we rather prefer to have a set of discrete emotions where our analysis follows established dictionary counting.
- In doing so, we have increased the theoretical justification for examining these specific emotions. We believe that the selected discrete emotions have theoretical backing that makes them valuable to investigate.
- Based on our pilot analysis, we filter for a subset of emotions that also have a notable correlation in the dictionary validation. In particular, anger, fear, joy and sadness had the strongest correlations within our user validation study among the 8 basic emotion categories, suggesting that these emotions are detectable to readers. This remedies also several of the concerns below.

- For breadth, we have retained the previous emotional analyses but moved them into the supplement. Therein, we are also transparent in that the validation revealed only a low correlation.

Comment R2.3: “The other concern I have is about the result of validation efforts. As I mentioned above, the correlation for discrete emotions is very weak. The authors also only note that the human judges are better than random chance for the topic intrusion test but not by how much. Better than random is a very weak baseline. How much better? How does that vary across topics? As I had mentioned in my original review, topic models are generally good at prevalent topics but can fail at rarer ones. I can imagine you can get better than random performance on average while having a very poor performance for many rarer topics. Overall, while there are validation efforts, it is unclear whether the bar of “good enough” has been set properly.”

Response: Thank you. We addressed your comments regarding the validation efforts in the following ways:

- Dictionary validation. By limiting the scope of our emotion analyses, our hypotheses are now derived only for discrete emotions that had a notable correlation with user judgements, namely anger, fear, joy, and sadness. Based on our pilot analysis, we discarded other emotions for which that was not the case (i.e., and did not use them in the preregistered hypotheses and not in the regression analysis).
- Topic validation. Regarding the topic models, users who completed the topic intrusion task correctly identified the topic of a given headline in 51.5% of trials. For comparison, a random guess would lead to 25% of the headlines being labeled correctly. Hence, twice as many headlines are assigned to the correct topic. This is significantly above chance, $\chi^2 = 249.61$, $P < 0.01$. These statistics can be found in Supplement H.3. Additionally, we investigated the rates of correct answers among the 7 topics. We have attached a table below, and included it in Supplement H.3. Six out of the seven categories are above chance, with only the category of *People* being close to chance rates.

Topic	Percent Correct
Entertainment	45.0%
Government & Economy	51.5%
LGTB	67.0%
Life	49.5%
Parenting & School	43.0%
People	25.3%
Women Rights & Feminism	76.0%

Comment R2.4: “A few other points are listed below:

Dictionary words: Some of the words in C.2 make me more worried. The top word for anger is words, the second most important for disgust is boy, top word for fear is watch.”

Response: The NRC was developed in a large-scale analysis where thousands of raters classified more than 10,000 terms according to the different emotions and thus presents the overall perception of a broader public (see *Crowdsourcing a Word-Emotion Association Lexicon*, Saif Mohammad and Peter Turney, *Computational Intelligence*, 29 (3), 436-465, 2013.). The NRC emotion lexicon is nowadays regarded as best-practice (Mohammad, Saif M. *Sentiment analysis: Detecting valence, emotions, and other affectual states from text*. *Emotion Measurement*. 201-237, (2016) and thus used in our study. However, the original paper acknowledges some ambiguity among classified emotions as raters perceived a word as a signal of such emotion while others did not (Mohammad, S. & Turney, P. *Emotions evoked by common words and phrases: Using mechanical turk to create an emotion lexicon*. In *Proceedings of the NAACL HLT 2010 Workshop on Computational Approaches to Analysis and Generation of Emotion in Text*, 26–34 (2010)). We acknowledge that the same uncertainty is also part of our paper, but argue that developing a better dictionary is beyond the scope of our work. In that, we are consistent with all others works that build upon the NRC emotion lexicon as best-practice (Mohammad, Saif M. *Sentiment analysis: Detecting valence, emotions, and other affectual states from text*. *Emotion Measurement*. 201-237, (2016)).

After reading your comment, we then manually inspected our dataset to get a better understanding of the above words. Here, it was often the case that the words appear in a context that is characterized by such emotion. For example, the term “boy” is often part of the expression

“Oh boy! ...” where it is used to signal strong opposition and even disgust. Similarly, the term “watch” was used in the context of “watch out” where, as a result, the headline was perceived as communicating fear.

Comment R2.5: “Dictionary validation: (section D)

You say: “The number of headlines (N = 213) and the number of raters were chosen based on best practices, laid out explicitly in [60].” Can you please write down your procedure explicitly? The readers shouldn't need to go to a different paper to know about your sampling design.”

Response: Thank you. We have updated the manuscript to make the above parts self-contained (rather than relegating to the literature). For this, we include more details about the sampling and validation procedure.

Comment R2.6: “You say you use spearman correlation but do not explicitly mention with what. Is it the average of 8 human judges' scores or median? Or did you look at each human judge score separately? I am hoping that it is not the latter, but I can't find a reference to this in the paper. These aggregation functions would have different implications, so please clarify.”

Response: We correlated the mean of the 8 human judges' scores for a headline with NRC sentiment rating for that headline. We have clarified this in the manuscript.

Comment R2.7: “Correlation of human judgment and dictionary approach: 0.3 is a weak correlation. Please note this in your writing.”

Response: Thank you. Following your suggestion, we noted this in our revised manuscript. Here, we followed psychological research, specifically Cohen's (1988) conventions to interpret effect size and call it a “moderate” correlation. We will also discuss this in the Discussion (however, due to the process of the Registered Report, we are only allowed to add a Discussion in Stage 2 and not in Stage 1).

Comment R2.8: “For discrete emotions, the correlation is even weaker. This, combined with the dictionary results from C2 and lack of theoretical grounding and interpretation of the related findings, makes me think this part of the paper needs to be strengthened or removed. For instance, given the weak correlation, did you look at the cases where human judgment disagrees with to see if there are some important patterns? You can also see that the correlation is not significant for some emotions.”

Response: We agree. We thus reorganized our proposed method and used the pilot analysis to filter for a subset of emotions that had a notable correlation with user judgements. These emotions were anger, fear, joy, and sadness. We removed the other emotions from our preregistered hypothesis, as well as from our proposed regression analysis.

We also agree that a more theoretical background for discrete emotions was needed. Therefore, we have strengthened this section in the several ways. As mentioned above, we now focus our emotional analysis on constructs that are theoretically backed and consistent with prior literature. Specifically, we only count the frequency of emotional terms. As part of this, we removed other measures from our paper (i.e., bipolar pairs, emotional dyads) for which we felt that the theoretical backing was not strong enough.

Response to Reviewer #3

Comment R3.1: "I am impressed by the thorough revision and the 'merged' effort to combine two interesting proposals into one study that certainly will provide interesting insights. As my main concern of lacking validation of the entertainment dictionaries has been rigorously addressed by the inclusion of several validation user studies, I believe the study proposal is now ready for stage 2."

Response: Thank you very much!

Decision Letter, fourth revision:

11th March 2022

Dear Stefan,

Thank you once again for submitting your revised Stage 1 Registered Report, entitled "Negativity drives online news consumption." Everything is in order and I am delighted to say that we can offer acceptance in principle. You may progress to Stage 2 and complete the study as approved.

We have just one small additional editorial request at this stage: we ask that you send an updated copy of your manuscript that includes a statement on the role of the funders in the Acknowledgments section.

That is, please indicate what role the funder(s) had in the conceptualization, design, data collection, analysis, decision to publish, or preparation of the manuscript. If any of this information could be perceived as a competing interest, ensure that it is also included in your competing interests statement. If the funder(s) had no role, please include the following statement: "The funders had no role in study design, data collection and analysis, decision to publish or preparation of the manuscript."

As you know, a condition of in-principle-acceptance is that the authors agree to deposit their Stage 1 accepted protocol in a repository, either publicly or under embargo until Stage 2 acceptance and publication. We are very keen to showcase our in-principle accepted protocols, so that our readers, reviewers, and potential authors can gain insight into the requirements of the format as well as an idea of the types of projects that are suitable for publication in Nature Human Behaviour. We have set up a space on figshare (https://springernature.figshare.com/registered-reports_NHB) to host all of our in-principle accepted protocols, which can either be made public or kept under embargo until Stage 2 acceptance (depending on author preference). This gives you the opportunity to have your work publicly associated with Nature Human Behaviour, and of course we will be very pleased to showcase your report if you agree to share it publicly.

Depositing the work on our figshare space does not preclude deposition of your Stage 1 protocol on other depositories – your protocol can also be posted on OSF, Dataverse, Dryad or any other public repository of your choice. You also do not need to do anything – if you agree with posting your protocol on our figshare space, we will upload your protocol on your behalf and either set it public or place it under embargo, depending on your choice. Your protocol will be licensed under a CC BY license (Creative Commons Attribution 4.0 International License). The CC BY license allows for maximum dissemination and re-use of open access materials and is preferred by many research funding bodies. Under this license users are free to share (copy, distribute and transmit) and remix (adapt) the contribution including for commercial purposes, providing they attribute the contribution in the manner specified by the author or licensor (read full legal code: <http://creativecommons.org/licenses/by/4.0/legalcode>) Please note that any use of <https://springernature.figshare.com> will be subject to the Figshare terms of use. Figshare has the right to enforce these terms and conditions where applicable. Use of third party services and sites will be subject to the relevant terms of use and will apply if we act on your behalf in this regard. Do let me know if you would like to take up this option or if you have any questions regarding the protocol deposition requirement.

Following completion of your study, we invite you to resubmit your paper for peer review as a Stage 2 Registered Report. Please note that your manuscript can still be rejected for publication at Stage 2 if the Editors consider any of the following to hold:

- The results were unable to test the authors' proposed hypotheses by failing to meet the approved outcome-neutral criteria
- The authors altered the Introduction, rationale, or hypotheses, as approved in the Stage 1 submission
- The authors failed to adhere closely to the registered experimental procedures
- Any post hoc (unregistered) analyses were either unjustified, insufficiently caveated, or overly dominant in shaping the authors' conclusions
- The authors' conclusions were not justified given the data obtained

We encourage you to read the complete guidelines for authors concerning Stage 2 submissions at <https://www.nature.com/nathumbehav/registeredreports>. Please especially note the requirements for protocol deposition, data sharing, and that withdrawing your manuscript will result in publication of a

Retracted Registration.

In recognition of the time and expertise our reviewers provide to Nature Human Behaviour's editorial process, we would like to formally acknowledge their contribution to the external peer review of your manuscript entitled "Negativity drives online news consumption". For those reviewers who give their assent, we will be publishing their names alongside the published article.

When you are ready, please use the following link to access your home page and submit your Stage 2 Registered Report:

[REDACTED]

*This url links to your confidential homepage and associated information about manuscripts you may have submitted or be reviewing for us. If you wish to forward this e-mail to co-authors, please delete this link to your homepage first.

We expect your Stage 2 Registered Report to be submitted according to the timeline specified in your latest cover letter. If unforeseen circumstances prevent submission by that date, please contact us as soon as possible to discuss any changes to the submission time-frame.

Thank you again for offering us this work and we look forward to receiving your Stage 2 Registered Report.

Yours sincerely,
Aisha

Aisha Bradshaw, PhD
Senior Editor
Nature Human Behaviour

Decision Letter, fifth revision:

1st August 2022

Dear Stefan,

Thank you once again for your Stage 2 Registered Report, entitled "Negativity drives online news consumption", and for your patience during the peer review process. I'm very sorry for the delay in reaching a decision.

Your manuscript has now been evaluated by Reviewers 1, 2, and 3 from the previous rounds of review. Reviewer 2 provided only remarks to the editors, indicating that they are satisfied with your Stage 2 manuscript. Comments from Reviewers 1 and 3 are included at the end of this letter. In the light of our reviewers' advice, we are pleased to inform you that we will be able accept your Stage 2 manuscript,

pending revisions to address reviewer comments and editorial requests.

To guide the scope of the revisions, the editors discuss the referee reports in detail within the team, including with the chief editor, with a view to (1) identifying key priorities that should be addressed in revision and (2) overruling referee requests that are beyond the scope of Stage 2 Registered Reports. We ask that you address the following points in your revision:

I. Reviewer 1 asks you to report skewness indicators for the main outcome measure, which we ask that you provide. Reviewer 1 also recommends additional robustness checks to address concerns about skewness in the main outcome measure. Per Registered Report policy, it is entirely up to the authors whether they undertake at Stage 2 additional exploratory analyses that are recommended by the reviewers. Although it is at your discretion whether to carry out the additional robustness checks, we feel that they would substantially strengthen confidence in the validity of your results and encourage you to undertake them. Should you add new robustness checks, please ensure that you make clear that they were not part of the original planned analyses.

II. Reviewer 1 raises some concerns about the interpretation of results in light of the effect sizes and pattern of results for specific emotions. Please carefully address these points as you revise through additional discussion and contextualization of the findings.

III. Please address Reviewer 1's comments regarding the characterization of existing literature. In keeping with our policy on novelty/priority claims, please also revise statements such as "Our research allows for unique insights" (pg. 21).

One of the main reasons for delays in eventual acceptance is failure to fully comply with editorial policies and formatting requirements. To assist you with finalizing your manuscript for publication, I attach a checklist that lists all of our editorial policies and formatting requirements.

Please attend to *every item* in the checklist and upload a copy of the completed checklist with your submission. I have listed below specific items that require your attention. I also mention here a few points that are frequently missed and can cause delays:

1) Ensure that the Stage 1 text does not change, except for minor tense changes. For instance, please remove the added statement "using N = 22,743 large-scale, randomized controlled trials in the field" from pg. 5. Around pg. 25, please restore the deleted statement "which corresponds to, on average, 4.27 headlines per experiment. On average, there were approximately 16,670 participants in each RCT".

2) Please ensure that the Abstract does not exceed 150 words. We request that you remove qualitative descriptors, such as "massive" and "one of the fastest growing websites of all time".

3) Please ensure that all statistical results are fully reported in-text, or include a reference to the relevant table where full results can be found. For instance, the results for text length, complexity score, and time trends discussed on pg. 17 should be fully reported.

4) When reporting supplementary results, please refer to specific tables, rather than sections of the Supplementary Information. For instance, on pg. 15, please refer to a specific table, rather than stating "Full results are in Supplement G".

- 5) Please include the number of observations for each figure panel in the figure captions.
- 6) At this stage, please provide a working link for the Code Availability statement.
- 7) Ensure that you provide all of the materials requested in the attached checklist and below with your final submission.

Nature Human Behaviour offers a transparent peer review option for new original research manuscripts submitted from 1st December 2019. We encourage increased transparency in peer review by publishing the reviewer comments, author rebuttal letters and editorial decision letters if the authors agree. Such peer review material is made available as a supplementary peer review file. **Please state in the cover letter 'I wish to participate in transparent peer review' if you want to opt in, or 'I do not wish to participate in transparent peer review' if you don't.** Failure to state your preference will result in delays in accepting your manuscript for publication.

Please note: we allow redactions to authors' rebuttal and reviewer comments in the interest of confidentiality. If you are concerned about the release of confidential data, please let us know specifically what information you would like to have removed. Please note that we cannot incorporate redactions for any other reasons. Reviewer names will be published in the peer review files if the reviewer signed the comments to authors, or if reviewers explicitly agree to release their name. For more information, please refer to our FAQ page.

We hope to hear from you within two months; please let us know if the revision process is likely to take longer.

To submit your revised manuscript, you will need to provide the following:

- Cover letter
- Point-by-point response to the reviewers (if applicable)
- Manuscript text (not including the figures) in .docx or .tex format
- Individual figure files (one figure per file)
- Extended Data & Supplementary Information, as instructed
- Reporting summary
- Editorial policy checklist
- Third-party rights table (if applicable)
- Suggestions for cover illustrations (if desired)

Consortia authorship:

For papers containing one or more consortia, all members of the consortium who contributed to the paper must be listed in the paper (i.e., print/online PDF). If necessary, individual authors can be listed in both the main author list and as a member of a consortium listed at the end of the paper. When submitting your revised manuscript via the online submission system, the consortium name should be entered as an author, together with the contact details of a nominated consortium representative. See <https://www.nature.com/authors/policies/authorship.html> for our authorship policy and <https://www.nature.com/documents/nr-consortia-formatting.pdf> for further consortia formatting guidelines, which should be adhered to prior to acceptance.

Forms:

Nature Human Behaviour has now transitioned to a unified Rights Collection system which will allow our Author Services team to quickly and easily collect the rights and permissions required to publish your work. Once your paper is accepted, you will receive an email in approximately 10 business days providing you with a link to complete the grant of rights. If you choose to publish Open Access, our Author Services team will also be in touch at that time regarding any additional information that may be required to arrange payment for your article.

[REDACTED]

With best regards,
Aisha

Aisha Bradshaw, PhD
Senior Editor
Nature Human Behaviour

Reviewer #1:

Remarks to the Author:

It was very interesting seeing the stage-2 paper submission. I have two suggestions concerning the assessment criteria given by Nature Human Behaviour.

1. Whether the data are able to test the authors' proposed hypotheses by passing the approved outcome-neural criteria

- As mentioned in previous review rounds, the data is impressive and original. Concerning data quality, however, I am a bit concerned about the skewness of the main outcome variable CTR. Can the authors provide any skewness indicators? Were any measures taken by the authors to circumvent potential statistical problems arising from this skewness (e.g., transformation, robust methods).

2. Whether the authors' conclusions are justified given the data

- Given the very small effects for the negativity bias, I think that the conclusions are a bit overstated. Is a 1-2% increase/decrease a meaningful difference? Can this be compared to previous or related studies? It would be helpful to contextualize the effect size.

- The discussion on the null effects for anger, and the opposite effects for fear need further elaboration.

These findings seem to be in opposite to theoretical expectations and previous studies. How can they then be explained? The authors conclude this paragraph by stating that "These findings extend prior research examining the effect of discrete emotions on online content sharing and message diffusion". However, do they really extend previous research or just contradict existing research, and if so, are there any explanations for this? If there are no theoretical explanations, this could hint at issues concerning the method. This finding is also interesting as it seems to hint not at a general negativity bias but a more specific one for specific negative emotional words. General conclusions about the negativity bias (i.e., "negative language leads to more clicks") might then not be fully correct, as this does not hold for words related to anger, and fear, for example.

- I do not agree with the authors' conclusion that they are the first to study negativity bias in news consumption, see for example:

Bachleda, S., Neuner, F. G., Soroka, S., Guggenheim, L., Fournier, P., & Naurin, E. (2020). Individual-level differences in negativity biases in news selection. *Personality and Individual Differences*, 155, 109675.

Trussler, M., & Soroka, S. (2014). Consumer demand for cynical and negative news frames. *The International Journal of Press/Politics*, 19(3), 360-379.

Knobloch-Westerwick, S., C. Mothes, and N. Polavin. 2020. "Confirmation Bias, Ingroup Bias, and Negativity Bias in Selective Exposure to Political Information." *Communication Research* 47 (1): 104-124.

Reviewer #2:

None

Reviewer #3:

Remarks to the Author:

Reviewing a preregistered report comes with the advantage of comparably short reviews for the second stage if the first stage was accepted in principle and the authors stuck to their analysis plan. As far as I can tell, all analyses follow the preregistered analysis plan; any deviations (including changes to the introduction and methods sections) are made transparent and well justified. The comprehensive supplementary materials underscore the methodological rigor of the authors. I concur with the conclusions drawn from the results, and especially the discussion of how Upworthy news (headlines) differ from other online news. Furthermore, I really like the cooperation of two research teams with similar project ideas as an outcome of the registered report format. As such, I see no reasons why the manuscript shouldn't be published in its current state.

Author Rebuttal, sixth revision

**Response letter NATHUMBEHAV-200611432F:
"Negativity drives online news consumption"**

Dear Dr. Bradshaw and Editors,

We are excited to hear that our paper has been accepted for publication at *Nature Human Behaviour* pending minor revisions. We thank the editorial team for their helpful feedback throughout the review process. Please find below an outline of the minor revisions we have made to the manuscript. We believe that these changes improved the quality of the manuscript and hope that you find them satisfactory.

Response to Editor Comments:

Comment E.0: "Thank you once again for your Stage 2 Registered Report, entitled "Negativity drives online news consumption", and for your patience during the peer review process. I'm very sorry for the delay in reaching a decision.

Your manuscript has now been evaluated by Reviewers 1, 2, and 3 from the previous rounds of review. Reviewer 2 provided only remarks to the editors, indicating that they are satisfied with your Stage 2 manuscript. Comments from Reviewers 1 and 3 are included at the end of this letter. In the light of our reviewers' advice, we are pleased to inform you that we will be able accept your Stage 2 manuscript, pending revisions to address reviewer comments and editorial requests."

Response: Thank you for the positive feedback! We have outlined the minor revisions we have made to the paper below.

Comment E.1: "To guide the scope of the revisions, the editors discuss the referee reports in detail within the team, including with the chief editor, with a view to (1) identifying key priorities that should be addressed in revision and (2) overruling referee requests that are beyond the scope of Stage 2 Registered Reports. We ask that you address the following points in your revision:

I. Reviewer 1 asks you to report skewness indicators for the main outcome measure, which we ask that you provide. Reviewer 1 also recommends additional robustness checks to address concerns about skewness in the main outcome measure. Per Registered Report policy, it is entirely up to the authors whether they undertake at Stage 2 additional exploratory analyses that are recommended by the reviewers. Although it is at your discretion whether to carry out the additional robustness checks, we feel that they would substantially strengthen confidence in the validity of your results and encourage you to undertake them. Should you add new robustness

checks, please ensure that you make clear that they were not part of the original planned analyses.”

Response: We have added a measure of skewness to the descriptive statistics in Supplement A. At your suggestion as well as Reviewer 1's, we have added a new robustness check in which we run a regression analysis using a log-transformed version of our outcome variable (i.e., of the CTR). We find nearly identical results to our preregistered analyses which has increased our confidence in our overall conclusions. The new analysis is in Supplement E.8, Table 11.

Comment E.2: “II. Reviewer 1 raises some concerns about the interpretation of results in light of the effect sizes and pattern of results for specific emotions. Please carefully address these points as you revise through additional discussion and contextualization of the findings.”

Response: We appreciate the feedback by Reviewer 1, which we carefully addressed in our discussion and the contextualization of the findings.

Comment E.3: “III. Please address Reviewer 1's comments regarding the characterization of existing literature. In keeping with our policy on novelty/priority claims, please also revise statements such as “Our research allows for unique insights” (pg. 21)”

Response: We have changed the language to recognize the existing literature. We have also removed wording around “novel” or “unique” in order to adhere to the editorial policy.

Comment E.4: “One of the main reasons for delays in eventual acceptance is failure to fully comply with editorial policies and formatting requirements. To assist you with finalizing your manuscript for publication, I attach a checklist that lists all of our editorial policies and formatting requirements.

*Please attend to *every item* in the checklist and upload a copy of the completed checklist with your submission. I have listed below specific items that require your attention. I also mention here a few points that are frequently missed and can cause delays:”*

Response: Thank you. We followed the editorial checklist (see attachment).

Comment E.5: “1) Ensure that the Stage 1 text does not change, except for minor tense changes.

For instance, please remove the added statement “using N = 22,743 large-scale, randomized controlled trials in the field” from pg. 5. Around pg. 25, please restore the deleted statement “which corresponds to, on average, 4.27 headlines per experiment. On average, there were approximately 16,670 participants in each RCT”.

Response: We have restored the deleted text and removed the additional text.

Comment E.6: “2) Please ensure that the Abstract does not exceed 150 words. We request that you remove qualitative descriptors, such as “massive” and “one of the fastest growing websites of all time”.

Response: We have updated the abstract to fit within the word count and removed qualitative descriptors.

Comment E.7: “3) Please ensure that all statistical results are fully reported in-text, or include a reference to the relevant table where full results can be found. For instance, the results for text length, complexity score, and time trends discussed on pg. 17 should be fully reported.”

Response: Thanks. The full results have now been added in the text.

Comment E.8: “4) When reporting supplementary results, please refer to specific tables, rather than sections of the Supplementary Information. For instance, on pg. 15, please refer to a specific table, rather than stating “Full results are in Supplement G”.

Response: We have now included the subsections and tables for any relevant references to the supplemental materials.

Comment E.9: “5) Please include the number of observations for each figure panel in the figure captions.”

Response: We added the number of observations as required for figures in our main manuscript.

Comment E.10” 6) At this stage, please provide a working link for the Code Availability statement.”

Response: We now have included a working link to our OSF repository that contains our code.

Comment E.11: “7) Ensure that you provide all of the materials requested in the attached checklist and below with your final submission.”

Response: Thanks for including this checklist – we have ensured that all necessary materials are included in our manuscript.

Response to Reviewer 1:

Comment R1.1: “It was very interesting seeing the stage-2 paper submission. I have two suggestions concerning the assessment criteria given by Nature Human Behaviour.”

Response: We are glad you found it interesting!

Comment R1.2: 1. “Whether the data are able to test the authors’ proposed hypotheses by passing the approved outcome-neural criteria

- As mentioned in previous review rounds, the data is impressive and original. Concerning data quality, however, I am a bit concerned about the skewness of the main outcome variable CTR. Can the authors provide any skewness indicators? Were any measures taken by the authors to circumvent potential statistical problems arising from this skewness (e.g., transformation, robust methods).”

Response: Thanks for bringing this up. To respond to these concerns, we have added a measure of skewness to the descriptive statistics in Supplement A. Further, we have added an additional supplemental analysis in which we run a regression analysis using a log-transformed version of our outcome variable (the CTR). We find nearly identical results to our pre registered analyses.

Comment R1.3: 2. “Whether the authors’ conclusions are justified given the data

- Given the very small effects for the negativity bias, I think that the conclusions are a bit overstated. Is a 1-2% increase/decrease a meaningful difference? Can this be compared to previous or related studies? It would be helpful to contextualize the effect size.”

Response: We extended our discussion to provide additional context on the effect sizes. As detailed in our discussion, it is important to distinguish between (public) sharing behavior and (private) news consumption. While the observed effect sizes were noticeably smaller than in studies analyzing *sharing behavior* (e.g., Berger & Schwartz, 2011), increases/decreases of 1-2% have been found to be meaningful differences when studying negativity bias in *news consumption* (Trussler & Saroka, 2014).

Comment R1.4: "The discussion on the null effects for anger, and the opposite effects for fear need further elaboration. These findings seem to be in opposite to theoretical expectations and previous studies. How can they then be explained? The authors conclude this paragraph by stating that "These findings extend prior research examining the effect of discrete emotions on online content sharing and message diffusion". However, do they really extend previous research or just contradict existing research, and if so, are there any explanations for this? If there are no theoretical explanations, this could hint at issues concerning the method. This finding is also interesting as it seems to hint not at a general negativity bias but a more specific one for specific negative emotional words. General conclusions about the negativity bias (i.e., "negative language leads to more clicks") might then not be fully correct, as this does not hold for words related to anger, and fear, for example."

Response: We agree that it is surprising that the results for anger and fear deviate from prior findings. We have now made it clearer in the manuscript that this may be the case due to the differences in our DV from prior studies. We now suggest that anger and fear may be more important when deciding whether to publicly share online content rather than privately consume it. We unfortunately cannot compare sharing and consumption behavior in this data set, but we hope to investigate it in the future, which we now mention.

Comment R1.5: "- I do not agree with the authors' conclusion that they are the first to study negativity bias in news consumption, see for example:

Bachleda, S., Neuner, F. G., Soroka, S., Guggenheim, L., Fournier, P., & Naurin, E. (2020). Individual-level differences in negativity biases in news selection. Personality and Individual Differences, 155, 109675.

Trussler, M., & Soroka, S. (2014). Consumer demand for cynical and negative news frames. The International Journal of Press/Politics, 19(3), 360-379.

Knobloch-Westerwick, S., C. Mothes, and N. Polavin. 2020. "Confirmation Bias, Ingroup Bias, and Negativity Bias in Selective Exposure to Political Information." *Communication Research* 47 (1): 104–124."

Response: Thanks for including these citations. We have changed the language to be more accurate in our manuscript and added relevant citations to our manuscript.

Response to Reviewer 3:

Comment R3.1: "Reviewing a preregistered report comes with the advantage of comparably short reviews for the second stage if the first stage was accepted in principle and the authors stuck to their analysis plan. As far as I can tell, all analyses follow the preregistered analysis plan; any deviations (including changes to the introduction and methods sections) are made transparent and well justified. The comprehensive supplementary materials underscore the methodological rigor of the authors. I concur with the conclusions drawn from the results, and especially the discussion of how Upworthy news (headlines) differ from other online news. Furthermore, I really like the cooperation of two research teams with similar project ideas as an outcome of the registered report format. As such, I see no reasons why the manuscript shouldn't be published in its current state."

Response: Thank you for the kind feedback!

Final Decision Letter:

Dear Stefan,

We are pleased to inform you that your Registered Report "Negativity drives online news consumption", has now been accepted for publication in *Nature Human Behaviour*.

Please note that *Nature Human Behaviour* is a Transformative Journal (TJ). Authors whose manuscript was first submitted on or after January 1st, 2021, may publish their research with us through the traditional subscription access route or make their paper immediately open access through payment of an article-processing charge (APC). Authors will not be required to make a final decision about access to their article until it has been accepted. IMPORTANT NOTE: Articles submitted before January 1st, 2021, are not eligible for Open Access publication. Find out more about Transformative Journals

Authors may need to take specific actions to achieve compliance with funder and institutional open access mandates. If your research is supported by a funder that requires immediate open access (e.g. according to Plan S principles) then you should select the gold OA route, and we will direct you to the compliant route where possible. For authors selecting the subscription

publication route, the journal's standard licensing terms will need to be accepted, including self-archiving policies. Those licensing terms will supersede any other terms that the author or any third party may assert apply to any version of the manuscript.

An online order form for reprints of your paper is available at www.nature.com/reprints/author-reprints.html. All co-authors, authors' institutions and authors' funding agencies can order reprints using the form appropriate to their geographical region.

With best regards,
Aisha

Aisha Bradshaw, PhD
Senior Editor
Nature Human Behaviour